



# Reviews and syntheses: Remotely sensed optical time series for monitoring vegetation productivity

Lammert Kooistra[1,*], Katja Berger[2,3,*], Benjamin Brede[4], Lukas Valentin Graf[5,6], Helge Aasen[5,6], Jean-Louis Roujean[7], Miriam Machwitz[8], Martin Schlerf[8], Clement Atzberger[9], Egor Prikaziuk[10], Dessislava Ganeva[11], Enrico Tomelleri[12], Holly Croft[13], Pablo Reyes Muñoz[2], Virginia Garcia Millan[14], Roshanak Darvishzadeh[10], Gerbrand Koren[15], Ittai Herrmann[16], Offer Rozenstein[17], Santiago Belda[18], Miina Rautiainen[19], Stein Rune Karlsen[20], Cláudio Figueira Silva[21], Sofia Cerasoli[21], Jon Pierre[3,22], Emine Tanır Kayıkçı[23], Andrej Halabuk[24], Esra Tunc Gormus[23], Frank Fluit[1], Zhanzhang Cai[25], Marlena Kycko[26], Thomas Udelhoven[27], and Jochem Verrelst[2]

[*]These authors contributed equally to this work.
[1]Wageningen University & Research, Laboratory of Geo-Information Science and Remote Sensing, Droevendaalsesteeg 3, 6708 PB Wageningen, the Netherlands
[2]Image Processing Laboratory (IPL), University of Valencia, C/Catedrático José Beltrán 2, Paterna, 46980 Valencia, Spain
[3]Mantle Labs GmbH, Vienna, Austria
[4]Helmholtz Center Potsdam GFZ German Research Centre for Geosciences, Section 1.4 Remote Sensing and Geoinformatics, Telegrafenberg, Potsdam, 14473, Germany
[5]Earth Observation of Agroecosystems Team, Division Agroecology and Environment, Agroscope, Zurich, Switzerland
[6]Institute of Agricultural Science, Crop Science, ETH Zürich, Zurich, Switzerland
[7]CESBIO, CNES, CNRS, INRAE, IRD, UT3, 18 avenue Edouard Belin, BPI 2801, TOULOUSE Cedex 9, 31401, France
[8]Remote Sensing and Natural Resources Modelling Group, Environmental Research and Innovation Department, Luxembourg Institute of Science and Technology (LIST), 41, rue du Brill, L-4422 Belvaux, Luxembourg
[9]Institute of Geomatics, University of Natural Resources and Life Sciences, Vienna (BOKU), Peter JordanStraße 82, 1190 Vienna, Austria
[10]Faculty Geo-Information Science and Earth Observation, ITC, University of Twente, The Netherlands
[11]Space Research and Technology Institute - Bulgarian Academy of Sciences, Georgi Bonchev bl. 1, 1113 Sofia, Bulgaria
[12]Faculty of Agricultural, Enviromental and Food Sciences; Free University of Bozen/Bolzano
[13]School of Biosciences, University of Sheffield, Sheffield, UK S10 2TN and Institute for Sustainable Food, University of Sheffield, Sheffield, UK S10 2TN
[14]European Topic Centre, University of Malaga, Arquitecto Francisco Peñalosa, 18, 29010 Málaga, Spain.
[15]Copernicus Institute of Sustainable Development, Utrecht University, Utrecht, The Netherlands
[16]The Plant Sensing Laboratory, The Robert H. Smith Institute for Plant Sciences and Genetics in Agriculture, The Hebrew University of Jerusalem, P.O. Box 12, Rehovot 7610001, Israel
[17]Institute of Soil, Water and Environmental Sciences, Agricultural Research Organization—Volcani Institute, HaMaccabim Road 68, P.O. Box 15159, Rishon LeZion 7528809, Israel
[18]UAVAC, Applied Mathematics Department, University of Alicante, 03080 Alicante, Spain
[19] School of Engineering, Department of Built Environment, Aalto University, Finland
[20]NORCE Norwegian Research Centre AS, P.O. Box 6434, 9294 Tromsø, Norway
[21]Forest Research Centre (CEF) and Associated Laboratory TERRA, School of Agriculture, University of Lisbon, Tapada da Ajuda, 1349-017 Lisbon, Portugal





[22]Geotree, Cardinal place, 100 Victoria street, London, SW1E 5JL, UK
[23]Karadeniz Technical University, Engineering Faculty, Department of Geomatics Engineering, Trabzon/Turkey
[24]Institute of Landscape Ecology, Slovak Academy of Sciences, 814 99 Bratislava, Slovakia
[25]Department of Physical Geography and Ecosystem Science, Lund University, Sölvegatan 12, S-223 62 Lund, Sweden
[26]Department of Geoinformatics, Cartography and Remote Sensing, Chair of Geomatics and Information Systems, Faculty of Geography and Regional Studies, University of Warsaw, Krakowskie Przedmieście 26/28, 00-927, Warszawa, Poland
[27]Trier University, EOCP - Earth Observation and Climate Processes, Environmental Remote Sensing & Geoinformatics, 54296 Trier

**Correspondence:** Katja Berger (katber@uv.es)

**Abstract.** Vegetation productivity is a critical indicator of global ecosystem health and is impacted by human activities and climate change. A wide range of optical sensing platforms, from ground-based to airborne and satellite, provide spatially continuous information on terrestrial vegetation status and functioning. As optical Earth observation (EO) data are usually routinely acquired, vegetation can be monitored repeatedly over time; reflecting seasonal

vegetation patterns and trends in vegetation productivity metrics. Such metrics include e.g., gross primary productivity, net primary productivity, biomass or yield. To summarize current knowledge, in this paper, we systematically reviewed time series (TS) literature for assessing state-of-the-art vegetation productivity monitoring approaches for different ecosystems based on optical remote sensing (RS) data. As the integration of solar-induced fluorescence (SIF) data in vegetation productivity processing chains has emerged as a promising source, we also include this

relatively recent sensor modality. We define three methodological categories to derive productivity metrics from remotely sensed TS of vegetation indices or quantitative traits: (i) trend analysis and anomaly detection, (ii) land surface phenology, and (iii) integration and assimilation of TS-derived metrics into statistical and process-based dynamic vegetation models (DVM). Although the majority of used TS data streams originate from data acquired from satellite platforms, TS data from aircraft and unoccupied aerial vehicles have found their way into produc-

tivity monitoring studies. To facilitate processing, we provide a list of common toolboxes for inferring productivity metrics and information from TS data. We further discuss validation strategies of the RS-data derived productivity metrics: (1) using *in situ* measured data, such as yield, (2) sensor networks of distinct sensors, including spectro-radiometers, flux towers, or phenological cameras, and (3) inter-comparison of different productivity products or modelled estimates. Finally, we address current challenges and propose a conceptual framework for productivity

metrics derivation, including fully-integrated DVMs and radiative transfer models here labelled as "Digital Twin". This novel framework meets the requirements of multiple ecosystems and enables both an improved understanding of vegetation temporal dynamics in response to climate and environmental drivers and also enhances the accuracy of vegetation productivity monitoring.





## 1 Introduction

Vegetation productivity is the rate at which solar energy is converted into biomass. This productivity is the origin of all fuel, fibre, and food by which humanity and many other species live, and should therefore be closely monitored. According to the United Nations (UN), the global population is expected to reach 9.7 billion by 2050; presenting a significant challenge for ensuring sufficient future food production. The productivity of plants is a crucial factor in meeting this challenge, as it directly affects the amount of food that can be produced. Plant productivity thus 30 fundamentally delineates the habitability of our planet (Running et al., 2000).

The intensification and spatial expansion of human activities in recent centuries have profoundly altered the world's natural and cultural landscapes (Winkler et al., 2021), and have had a significant impact on ecosystem processes, and their functions in society. An integrative proxy of this global change is the altered regime of vegetation productivity.

As a key characteristic of ecosystem conditions, global vegetation productivity reflects both, the spatial distribu-35 tion and change of the vegetation coverage (EEA, 2021). The key climatic drivers of vegetation productivity are temperature, water supply and solar radiation (Madani et al., 2018), which interact to constrain the magnitude and temporal dynamics of ecosystem productivity depending on soil conditions. In the twenty-first century, it is expected that vegetation productivity will decrease due to the impacts of climate change in the Northern Hemisphere, and may negatively affect the global land carbon (C) sink with unknown feedback effects (Zhang et al., 2022b).

Optical remote sensing (RS) allows the monitoring and quantification of vegetation productivity from a local to a global scale. Consequently, among the objectives of the Earth observation (EO) satellite missions launched in the last five decades, primary importance has been given to observing the productivity and health of natural and cultivated vegetation land covers (e.g., Chevrel et al., 1981; Huete et al., 2002; Zhang et al., 2003; Atzberger, 2013). Novel satellite systems are launched constantly and significant improvements in data-driven, as well as physically 45 based data analysis techniques are made (Baret and Buis, 2008). These developments demand a systematic overview of the state-of-the-art in productivity related time series (TS) studies, presenting the unprecedented availability of continuous multi-sensor data streams, the constantly updated data repositories, and the latest processing techniques and toolboxes. Recent review papers focused on global land surface phenology (LSP) research (Zeng et al., 2020; Caparros-Santiago et al., 2021), but lacked the relationship to vegetation productivity. Other reviews were restricted 50 to specific ecosystems (Berra and Gaulton, 2021) or sensors (Eitel et al., 2016). Microwave-based studies were covered by Teubner et al. (2018); Wild et al. (2022). Our study aims to complement and extend previous efforts by providing a systematic review of the assessment of vegetation productivity using remotely sensed TS data streams. Consequently, we will focus on the literature that uses remotely sensed optical TS and derived proxies for quantifying productivity.

The review is divided into seven sections, where Sect. 1 provides a tangible definition of productivity and its temporal variability, and also introduces main productivity metrics. Sect. 2 describes the available optical sensor platforms. Sect. 3 provides the methods and toolboxes for processing, analysing and modelling time series. In Sect. 4

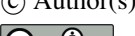



we outline three different strategies for validating productivity products. The outcomes of the systematic literature review are provided in Sect. 5. In Sect. 6 we provide an outlook on future challenges to assess vegetation productivity from TS data, followed by a conclusion (Sect. 7).

## 1.1 Definition of productivity adopted for this review

Productivity in ecosystems quantifies the rate at which autotrophic organisms, such as green plants, convert energy into organic metabolic assimilates (Scurlock and Olson, 2002; Larcher, 2003). Vegetation productivity is commonly defined in three measures: gross primary productivity (GPP), net primary productivity (NPP) and net ecosystem productivity (NEP). The interplay of these main productivity measures is illustrated in Fig. 1.

Over small spatial extents ($< 1\text{km}^2$), NEP is usually directly estimated through eddy-covariance (EC) methods, where the vertical, turbulent fluxes of $CO_2$ are measured within the atmospheric boundary layer using $CO_2$ concentration measurements from an infrared gas analyzer (IRGA) along with high-frequency sonic anemometer wind velocity measurements. NEP is subsequently partitioned into GPP and ecosystem respiration (Re), where estimated Re values are commonly derived from nighttime fluxes (i.e. NEP = Re) when GPP is zero and extrapolated to daytime fluxes. The ratio of NPP to GPP is termed the carbon use efficiency and represents the capacity to which plants are able to transform assimilated $CO_2$ into stored biomass, after carbon losses through autotrophic respiration (Ra). The carbon use efficiency of vegetation varies according to factors such as plant species, nutrient availability, light, temperature and water availability. However, the ratio of NPP to GPP is typically thought to be around 0.45, according to empirical studies, satellite products and process-based models (He et al., 2018), indicating that 55% of the carbon captured by plants is directed towards respiration and thus cannot be utilized for net production and growth (Field et al., 1998).

Although NPP and GPP are common metrics to express the productivity of any ecosystem, in the literature different definitions or terms can be found. For instance, in agroecosystems, productivity often refers to aboveground (and below-ground) biomass (AGB) and yield (Chopping et al., 2011; Mariotto et al., 2013). In forestry, productivity is also often related to AGB or harvestable wood (Battles, 2022). For natural ecosystems, AGB, (e.g., Ramoelo et al., 2015; Lumbierres et al., 2017) but also directly NPP and GPP are commonly used to express productivity (see e.g. reviews by Anav et al. 2015; Liao et al. 2023). In the current review, we refer to all these productivity metrics, which are summarized in the blue box.





---

**Definitions of productivity metrics**

Productivity is the rate at which a quantity (e.g., energy) is accumulated by producers (e.g., plants) over time within a given area. Here we give an overview of metrics for productivity adopted in our review:

- **Gross primary productivity (GPP)** is the total amount of C photosynthesized by plants (Myneni et al., 1995) in a given time (g C/m$^2$/day) and describes also the largest carbon flux between the biosphere and the atmosphere (approximately 130 Gt C per year) (Krause et al., 2022).

- **Net primary productivity (NPP)** denotes the remaining C from photosynthesis after respiration losses from plants (Ra) (g C/m$^2$/day), which is invested for the maintenance of cells and the growth of tissues (Roxburgh et al., 2005).

- **Net ecosystem productivity (NEP)** is defined as NPP minus soil heterotrophic respiration (Rh) by microorganisms (g C/m$^2$/day) (Landsberg and Gower, 1997) (i.e., C loss from the decomposition of woody detritus, soil organic matter, vegetation mortality, grazing etc.). It reflects the temporal change in C that can be stored in an ecosystem (Harmon et al., 2011). NEP thus quantifies the loss or accumulation of C within an ecosystem and defines if it is acting as a sink or source of C.

- **Aboveground biomass (AGB)** is the total amount of plant matter on the soil surface in a given area or ecosystem that has accumulated over time, as a result of photosynthesis and plant metabolism (kg C/m$^2$). AGB plays a crucial role in quantifying the productivity of forests etc. as it specifies the amount of stored carbon per unit area and subsequently the capacity for water filtration, soil retention, and biodiversity conservation (Powell et al., 2010; Goetz et al., 2009).

- **Crop yield** is defined as the amount of the harvested product (e.g., kg grain) per unit cropped area (kg/ha), and is a measure of productivity referring to the part of biomass that can be used for the nutrition of humans, feeding of livestock, the production of fuel or construction materials (Carletto et al., 2015).

- **Harvestable wood** refers to productivity in forests, typically given in cubic meters of harvestable wood grown per year on a forested site (m$^3$/ha) (FAO, 2010).


## 1.2 Time series observations for productivity studies

Vegetation productivity is controlled by two processes; (i) the assimilation of $CO_2$ substrate through photosynthesis (source activity) and (ii) tissue growth from the accumulated carbohydrates into stored biomass (sink activity) (Körner, 2015). Plant photosynthesis is driven by incoming photosynthetic active radiation, $CO_2$ concentrations, 90 temperature, and water and nutrient availability. However, the actual growth of tissues from assimilated carbohy-


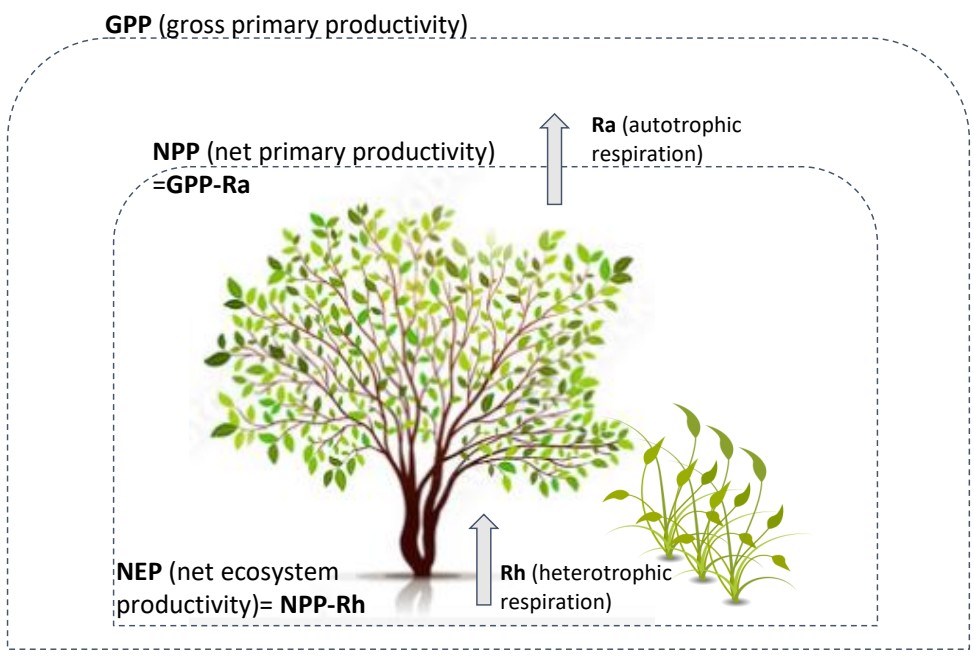

**Figure 1.** Distinction and interplay between GPP, NPP, and NEP, with autotrophic (plants) and heterotrophic respiration. Inspired by Mancini et al. (2016).

drates is also affected by other mechanistic processes that limit growth under restrictive temperature, moisture or nutrient conditions (Körner, 2015). Overall productivity therefore varies over time, across diurnal, seasonal, intra-annual and inter-annual timescales, and has a strong spatial component, due to variations in vegetation composition, soil properties and management practices.

To achieve accurate global estimates of plant productivity, and to understand the importance of climatic and environmental drivers, it is vital that the temporal data streams are paired with spatially-continuous observations through a fleet of optical EO satellites. In this manner, the metabolic pulse of terrestrial vegetation can be monitored. Data acquired from stationary or lower altitude platforms such as flux towers (e.g., Gamon, 2015), phenological cameras (phenocams) (e.g., Aasen et al., 2020) or Unoccupied Aerial Vehicles (UAV) permit a local "zoom-in" along
temporal and/or spatial dimensions. At the local to landscape scales, tower-mounted sensors and UAV platforms may provide even diurnal variations of fluxes (i.e., carbon). Each of these platforms provides sequences of recordings that enable together routine sensing of local-to-global vegetation productivity leading to an unprecedented influx of data streams (Sagan et al., 2019; Alvarez-Vanhard et al., 2021). Sect. 2 will discuss the most relevant platforms for remote sensing of productivity metrics.





Remotely sensed TS also capture the changes in the timing of recurrent, annual biological events (e.g. budburst, flower blossoming, leaf senescence). These phenological events are unequivocally related to productivity (e.g., Zhu et al., 2016), and shifts in the timings of seasonal phenological events have been shown to be related to inter-annual variations in annual GPP (e.g., Park et al., 2019). The extraction of land surface phenology LSP metrics (e.g., the start of the season or length of the growing season) can be done by fitting models to near-continuous

TS data (De Beurs and Henebry, 2004). However, TS data streams are not free from shortcomings (Meroni et al., 2019). In reality, the availability of continuous data is often hampered by: (1) sub-optimum to inadequate weather conditions, such as clouds, snow, dust and aerosols (e.g., Kandasamy et al., 2013) or (2) instrumentation errors and uncertainties (Graf et al., 2023), as well as calibration issues (e.g., Brinckmann et al., 2013). Cloud cover is the most stringent limitation of optical satellite data: the majority of the terrestrial Earth's surface is more or less regularly

covered by clouds, and for some areas, persistent cloud cover can last for weeks (e.g., Atkinson et al., 2012; Wilson and Jetz, 2016). In such cases, data sparsity leads to biased estimates. For instance, clouds can mask key stages of phenological events, leading to unreliable monitoring practices such as productivity predictions (e.g., Karlsen et al., 2018). The spatiotemporal gap-filling of missing TS data has therefore become a crucial step for monitoring the life cycle of vegetation and inter- and intra-annual variations in plant productivity (e.g., Beck et al., 2006; Schwartz,

2013; Amin et al., 2022). See also the discussion in Sect. 3.

### 1.3    Measuring productivity with optical Earth observation data

The presence of strong absorption features in optical wavelengths, which relate to biochemical properties such as pigment and water content, has led to a large body of research using optical sensors to monitor vegetation productivity (e.g., Boisvenue et al., 2016; Brinkmann et al., 2011; Cai et al., 2021; Dusseux et al., 2022; Erasmi

et al., 2021; Hill and Donald, 2003). Given the employment of optical sensors routinely recording data at different scales, generated data streams have gradually become a well-established source of information in a wide array of vegetation monitoring applications, such as assessing climate change impact and carbon modelling (e.g., Campbell et al., 2022; Wocher et al., 2022), drought monitoring (e.g., Atzberger et al., 2013) or biodiversity assessment (e.g., Lausch et al., 2020).

Traditionally, spectral vegetation indices (VIs) have been used as proxies for plant productivity or stored biomass (e.g., Erasmi et al., 2021; Fiore et al., 2020), with other studies focusing on dynamically integrating vegetation traits within more complex data-driven and process-based models (e.g., Ardö, 2015; Pei et al., 2022). Light use efficiency (LUE) schemes (Monteith, 1972) model GPP as a function of the amount of incoming photosynthetically active radiation (PAR) and the fraction of absorbed PAR (fAPAR) along with an LUE term (e.g., Zhao et al., 2005; Wang et al.,

2017). See also the extensive review by Pei et al. (2022) and seminal papers by Moulin et al. (1998) and Delécolle et al. (1992). VIs that are sensitive to fAPAR and related vegetation traits (e.g. chlorophyll content, leaf area index: LAI) have been integrated into LUE-based approaches, to represent physiological constraints on GPP (e.g., Gitelson et al., 2003; Cheng et al., 2014b; Xie et al., 2019). Data-driven approaches may include the establishment of statisti-





cal relationships through empirical approaches, or more recently with machine learning (ML) algorithms (see review
by Liao et al. 2023). Over the last decade, solar-induced fluorescence (SIF) became increasingly popular, giving a
more direct measure of photosynthetic activity, and thus serves as perhaps the most straightforward remotely sensed
proxy for GPP (e.g., Frankenberg et al., 2011; Guanter et al., 2012). In the most complex modeling approaches,
GPP is inferred using process-based dynamic vegetation models (DVM) (e.g., Krinner et al., 2005; Sitch et al.,
2003; Liu et al., 2014). DVMs can be both diagnostic and prognostic tools, able to simulate responses to climatic
change including prognoses of carbon budgets (e.g., Rayner et al., 2005; Ardö, 2015). Overall, the development of
methodologies is further accelerated by a vast increase and the long-term vision of EO data, the availability of
historical data, together with enhanced facilities through numerous data repositories. Subsequently, data analytics
and data-driven ML methods have helped the spread and penetration of these (big) data into data-based services
worldwide (Liu, 2015; Gorelick et al., 2017). The methodologies are discussed in more detail in Sect. 3.

## 2  Sensor platforms for vegetation productivity monitoring

Over the last two decades, the optical EO domain has seen an increasing number of space missions with various sensors
aboard, complemented by airborne campaigns and *in situ* measurements from widespread ground-based networks.
This increase in the abundance of EO data has contributed to the establishment of consistent global databases with
quality-checked optical data, which can be used to estimate vegetation productivity metrics at almost any spatial
and temporal scales (Kuenzer et al., 2015). The relevant sensor platforms serving to collect observations for deriving
vegetation productivity information are graphically illustrated in Fig. 2 and described in the following sections.

### 2.1  Time series from EO satellites

In recent years, the availability of free satellite data has dramatically increased, amounting to petabytes of data. This
expansion is due to the decreasing costs of data acquisition and the constant reduction of required computational
resources and storage infrastructure. The review by Ustin and Middleton (2021) provides a detailed description of
this trend. The availability of such data reinforces the usefulness of satellite data streams for capturing vegetation
dynamics at various spatial scales, from monitoring local ecological habitats to conducting global studies (Cavender-
Bares et al., 2020). Fig. 3 summarizes the available optical (main) sensors starting from the 1970s with their spatial
resolution and revisit time.

Low Elevation Orbit (LEO) satellites have aboard sensors scanning at moderate (i.e., hecto- to kilo-metric)
spatial resolutions, such as the Advanced Very High-Resolution Radiometer (AVHRR), Moderate Resolution Imaging
Spectroradiometer (MODIS), Visible Infrared Imaging Radiometer Suite (VIIRS), PROBA-V (Project for On-Board
Autonomy - Vegetation), and Ocean and Land Colour Instrument (OLCI) onboard Sentinel-3. They provide high
frequency and long-term TS and thus support a deep investigation of the land surface phenology and trends along
with a thorough monitoring of vegetation productivity of the entire Earth (see reviews by Zeng et al. (2020);





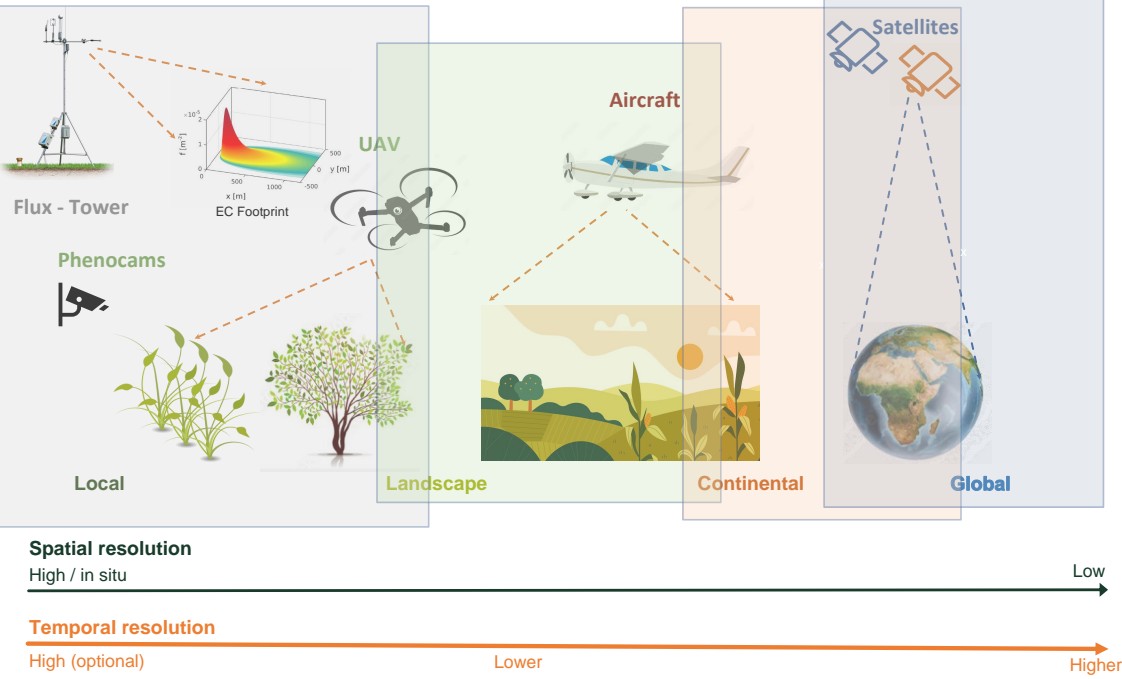

**Figure 2.** Overview of near and RS platforms used for vegetation productivity related TS analysis, i.e., flux towers with an exemplary footprint, phenocams, UAVs, aircraft and satellites. The platforms are arranged in order from left to right, starting with the highest spatial resolution and progressively decreasing (i.e., from high to low), although EC footprint sizes may vary. In terms of temporal resolution, the leftmost platforms, i.e., phenocams, typically offer higher optional temporal resolutions. Moving towards the right, the temporal resolution decreases (e.g., with aircraft platforms), and then it increases again as we transition towards EO satellite platforms. Figure elements are own creations, except for the flux tower and EC footprint (Kljun et al., 2015).

Pipia et al. (2022a)). On the other hand, Geostationary Earth Observation (GEO) satellites offer an opportunity to capture rapid changes in vegetation dynamics thanks to their high revisit frequency, spanning over a few minutes. For instance, mapping vegetation on an hourly basis by means of GEO satellites was explored by Fensholt et al. (2006), using the SEVIRI instrument on board Meteosat Second Generation (MSG). Another mission of interest is NASA's

Earth Polychromatic Imaging Camera (EPIC) onboard NOAA's Deep Space Climate Observatory (DSCOVR) (Yang et al., 2017). The EPIC team's primary responsibility is to develop and validate algorithms that produce a series of products, including the vegetation green LAI (GLAI) and its sunlit portion, at a spatial resolution of 10 km. GLAI and its sunlit portion are critical state parameters in many ecosystem productivity models (e.g., Bonan et al., 2003; Bi et al., 2022).





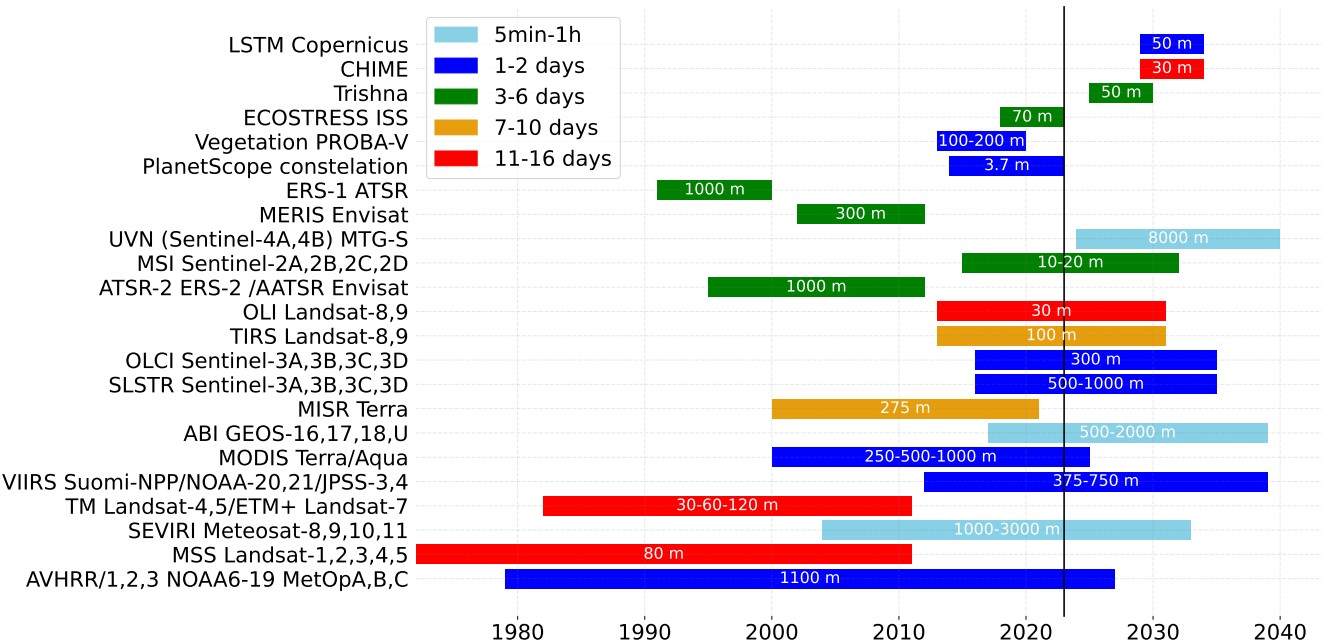

**Figure 3.** Optical EO sensors providing TS data starting from the 1970s. Different colours indicate the revisit time, and the spatial resolution for each sensor is given within the corresponding bars.

Launched in 2018, the ECOSTRESS mission onboard the International Space Station (ISS) delivers nominally daily land surface temperature (LST) products in taking advantage of the fast orbiting ISS (note that the real revisit period for a given location is variable and depends on the instrument's orbital cycle aboard the ISS). The spatial resolution of the products is 70 m except for two products of 30 m, due to the low altitude (Li et al., 2021b). In addition to the opening of the Landsat archives in 2008, further momentum was gained through the European Sentinel missions (Berger et al., 2012). From 2015 onward, Sentinel-2 (S2) optical imagery has been offering unprecedented perspectives on the temporal variability of plant productivity of different ecosystems, e.g. grasslands (Dusseux et al., 2022) or forests (Lin et al., 2019) and its divergence over fine spatial scales. Compared to earlier land satellite missions, such as MODIS or Landsat, S2 provides improvements in revisit time (5 days at the equator), spatial resolution (10-20 m) and spectral configuration (more and narrower vegetation-related bands) (Drusch et al., 2012). A 5-day revisit time may still pose limitations in acquiring a satisfactory number of cloud-free scenes required to construct a comprehensive composite product for productivity modelling in a dynamic ecosystem. This constraint becomes particularly crucial during transitional phases such as bud-burst and senescence, as well as (a)biotic stress events or following rainfall in water-limited ecosystems like drylands. It is worth mentioning that the near-polar orbit of S2 allows for a higher number of acquisitions when approaching the poles. For instance, over the high arctic archipelago Svalbard, S2 images can be obtained twice a day, allowing for regional-scale mapping of plant





productivity via LSP metrics (Karlsen et al., 2021). For continental Europe, continuous phenological mapping using S2 is today operational in the Copernicus pan-European High-Resolution Vegetation Phenology and Productivity product suite (HR-VPP) project (Tian et al., 2021). The exploitation of Sentinel-3 OLCI data even ensures a daily global coverage although at a moderate spatial resolution (300 m), but with a higher number of bands, allowing the derivation of essential vegetation traits for productivity monitoring studies (e.g., Yang et al., 2021b; Reyes-Muñoz et al., 2022).

Until recently, a fine temporal revisit time was at the expense of fine spatial resolution. However, a new generation of satellite constellations is breaking these formerly restrictive inter-dependencies (see Fig. 3) with, for instance, the constellation of PlanetScope satellites that offer multispectral images at 3 m spatial resolution in daily revisit intervals (Roy et al., 2021).

## 2.2 Time series from piloted aircraft and unoccupied aerial vehicles

Aircraft constitute flexible and adaptable platforms to explore new protocols of measurements, support applied studies (e.g., Cheng et al., 2014a; Atzberger et al., 2015), and perform CalVal activities. However, in contrast to orbital platforms, the regular acquisition of TS using an aircraft is a logistic and financial burden. This may explain why we could not identify studies that employed piloted aircraft to acquire optical TS for the estimation of vegetation productivity. To capture time trends at a smaller patch scale, UAVs have emerged as a more efficient and cheaper option. Theoretically, UAVs meet most requirements for TS acquisitions regarding covering high spatial, spectral, and temporal resolutions (Berni et al., 2009; Aasen et al., 2018). UAVs are flexible and in contrast to satellite systems may be deployed whenever weather conditions are favourable for a desired measurement. Also, UAVs offer the necessary flexibility to sample diurnal cycles. In the research area of field phenotyping, UAVs have become a standard tool to capture high-resolution TS of plant growth (Aasen and Roth, 2022). To date, a range of multi-spectral and a few science-grade hyperspectral sensors have become available on the commercial market (Aasen et al., 2018), allowing for even faster system integration. In terms of TS analysis for productivity, so far UAV measurements were mainly employed to fill gaps in satellite observations caused by cloudiness or sparse data (Dash et al., 2018; Alvarez-Vanhard et al., 2021). A recent phenotyping UAV study, however, collected UAV data of a soybean field trial at an unprecedented temporal resolution (Borra-Serrano et al., 2020) which allowed fitting growth curves with high accuracy (>90%) to derive relevant traits.

## 2.3 Multi-sensor and multi-scale synergies for time series

As data from different platforms and sensor modalities provide complementary information in terms of spatial, spectral and temporal domains, the fusion of RS observations is increasingly coming into focus. For example, in the review study by Berger et al. (2022), the synergistic usage of multiple optical spectral domains was described to detect the stress of crops. Since productivity is affected by crop stress, improved stress detection and monitoring would also help in productivity studies. While biotic and abiotic stressors can only be disentangled through a synergistic multi-





sensor usage, in productivity studies this synergy may be less relevant. Instead, multi-scale approaches, for instance
by combining spectral information from aircraft, UAVs and EO satellites (as described above) are more relevant.
In this way, advantages of at least two platform types can be explored, such as more frequent availability or higher
spatial resolution data (e.g., Gevaert et al., 2015; Sagan et al., 2019; Alvarez-Vanhard et al., 2021). By providing
a higher number of observations, multi-sensor fusion improves the spatiotemporal continuity through gap-filling,
leading to higher consistency and accuracy of current satellite products related to vegetation productivity (e.g.,
Claverie et al., 2018; Manivasagam et al., 2019; Sadeh et al., 2021). Although not explicitly treated in this review,
additionally, the fusion of synthetic aperture radar (SAR) and optical TS data can be beneficial for productivity
monitoring in regions with frequent cloud coverage (e.g., Pipia et al., 2019; Mercier et al., 2020; Caballero et al.,
2023).

## 2.4   Service platforms

For the extraction of multi-year TS of satellite data, the concept known as "Platform as a Service" (PaaS) is a
recent development, allowing users to create TS for a region of interest (ROI) using scalable computing platforms
without having to download the images required. This "zero-download" paradigm is implemented, e.g., in the widely
used Google Earth Engine (GEE) (Gorelick et al., 2017). In addition to proprietary platforms, approaches based
on the Open Data Cube (ODC) initiative (Killough, 2018) offer the possibility of scalable extraction of spectral
satellite data over long periods of time. Another emerging cloud platform is the European openEo initiative. The
openEO API aims to standardise the interface between frontend users and backend platforms to process big EO
data (Schramm et al., 2021). An implementation of this cloud technology is the Digital Earth (DE) Africa (DE
Africa, 2022) platform. DE Africa allows the extraction and analysis of vegetation TS from Landsat, Sentinel-1 and
S2 data at the continental level as a web service running in a sandbox.
Another framework that works in a pure open-source manner is the Earth Observation Data Analysis Library
(EOdal, Graf et al. 2022). EOdal allows the intersection of EO data with environmental covariates that control
vegetation productivity using lightweight Python programming. The software can be used on local premises as well
as in cloud environments, making it a powerful open-source alternative to proprietary solutions while being far less
complex than, for example, ODC. Several national initiatives aim to establish an open EO data infrastructure, for
example, those of Austria, Australia, Switzerland, and Tanzania, facilitating large-scale environmental monitoring
and policy enforcement (Dhu et al., 2019).
Alternatively, the Forestry Thematic Exploitation Platform (F-TEP) among other TEPs such as the Foodsecurity-
TEP (FS-TEP) and the Coastel-TEP (C-TEP) provide easy access to satellite data worldwide. F-TEP offers online
processing services and tools to generate forest information products, and users can create and share their own
services and products. The platform includes pre-processed optical and radar data from Sentinel satellites and
other instruments, as well as additional data. Push-button functionalities are available for simple products such as
vegetation indices, while programmed services include forest and land cover maps, change maps, and continuous





forest variables which can be used to derive productivity metrics. Many of the other TEPs provide similar data and functionalities for their thematic domain.

## 3 Time series processing methods for vegetation productivity monitoring

### 3.1 Time series sources and pre-processing

#### 3.1.1 Vegetation Indices

VIs are widely applied methods for monitoring trends and plant productivity (e.g., Gutman, 1999; Huete et al., 2002; Atzberger and Eilers, 2011a; Rasmussen et al., 2014; Kang et al., 2018; Zeng et al., 2020; Shammi and Meng, 2021). Certainly, the most widely used VI in EO observation TS analysis is the normalized difference vegetation index (NDVI) (Rouse et al., 1974; Tucker, 1979). Its popularity comes from the fact that NDVI explores the contrasting behaviour of reflectance in the visible red and the near-infrared (NIR) spectral domains, which strongly relate to vegetation biomass, and by extension canopy-level plant photosynthetic activity. NDVI has the great benefit of being available to the research community through long observational records of more than five decades, specifically from AVHRR, the Landsat series, and MODIS (e.g., Huang et al., 2021; Li et al., 2021a). In addition to NDVI, other VIs have also been used to model temporal variations in productivity, including the enhanced vegetation index (EVI), which also accounts for canopy background and atmospheric influences (Huete et al., 2002). Multiple studies have explored TS of NDVI and EVI with direct linkages to vegetation productivity (e.g., Shi et al., 2017; Shammi and Meng, 2021), or as part of GPP assimilation schemes (e.g., Zhang et al., 2015; Liu et al., 2021a). However, biomass-sensitive VIs often overestimate GPP at the start and end of the growing season, when leaf chlorophyll content decouples from LAI (Croft et al., 2014, 2015). Recently, novel VIs have been proposed for time series analysis, such as the Plant Phenology Index (PPI, Jin and Eklundh 2014), which is used for the calculation of the High Resolution Phenology and Productivity (HR-VPP) product at 10 m resolution as part of the Copernicus Land Monitoring Service (Tian et al., 2021).

Despite their widespread usage, VIs also suffer from several drawbacks. Reducing the spectral signals into simple indices intrinsically leads to remaining spectral information unexploited, which potentially could inform about plant physiology (e.g., Atzberger et al., 2011; Verrelst et al., 2019a). In general, these parametric methods neglect the effect of the background soil and other confounding factors (e.g., Darvishzadeh et al., 2008; Verrelst et al., 2008, 2010; Gao et al., 2022). Also, they tend to be proxies for a small set of the physiological properties of vegetation only, leaving their empirical biophysical and biochemical meaning often ambiguous (e.g., Myneni et al., 1995; Morcillo-Pallarés et al., 2019).





### 3.1.2 Quantitative traits

A more explicit cause-effect alternative to obtaining TS of VIs can be derived from the radiative transfer theory. Radiative transfer models (RTMs) offer the possibility of deriving biochemical and biophysical traits at leaf (e.g.,

Jacquemoud et al., 1996; Ceccato et al., 2001; Féret et al., 2017) and canopy levels (e.g., Myneni et al., 1997; Rautiainen, 2005; Richter et al., 2009; Darvishzadeh et al., 2011) from optical remotely sensed data. RTMs describe the relationship between biochemical and biophysical traits and plant optical properties based on physical laws. Various inversion strategies have been developed based on lookup tables, numerical optimization methods, and ML methods, i.e. so-called hybrid approaches. An overview of RTM-based retrieval methods is provided by the reviews

of Kimes et al. (1998); Baret and Buis (2008); Verrelst et al. (2015a, 2019a). Building upon these RTM inversion strategies, a few traits are operationally retrieved from routinely acquired EO data from land missions such as MODIS or S2. The most widely produced vegetation products are LAI and fAPAR, but also fractional vegetation cover, and to a lesser extent canopy chlorophyll content (e.g., Myneni et al., 2015; Yan et al., 2016; Fang et al., 2019; Xu et al., 2022). In turn, TS data streams of these products have been integrated into various GPP assimilation

schemes (e.g., Jung et al., 2007; Xie et al., 2019; Chen et al., 2022).

Apart from those routinely generated vegetation products, a wide range of experimental studies present alternative retrieval methods or focused on the retrieval of other biochemical traits, e.g. leaf and canopy water content, leaf chlorophyll content (e.g., Croft et al., 2020; Esté et al., 2021; Caballero et al., 2023). Typically, these studies have been limited to the processing of single-date observations or at best multi-temporal acquisitions for a restricted time

window. Given those experimental retrievals, efforts to provide TS of a range of biochemical and biophysical traits were conducted by a few studies (e.g., Verger et al., 2016; Salinero-Delgado and Verrelst, 2021). An important note on the use of RTMs to derive quantitative traits concerns their sensitivity to phenological developmental stages of vegetation: Schiefer et al. (2021) demonstrated that trait retrieval accuracy has a strong dependency on phenology. A possible solution would be to use expert knowledge and *in situ* data to enable a more precise parameterisation

of the RTMs depending on the phenological (macro) phase. At the same time, fast processing speeds are required to retrieve traits from TS data streams. This points towards hybrid retrieval schemes including active learning, i.e., relying on tuning RTM simulations against *in situ* measured traits and training of ML algorithms (e.g., Verrelst et al., 2021; Berger et al., 2021).

An overview of widely used quantitative traits in TS processing available from RTM inversion, and their rela-

tionship to potential vegetation productivity is given in Table 1. These include proxies for photosynthetically active foliage area and mechanistic proxies for photosynthesis directly related to GPP. Especially the capability of advanced RTMs such as SCOPE (Soil Canopy Observation, Photochemistry and Energy fluxes, (Van der Tol et al., 2009; Yang et al., 2021a)) to model SIF is promising. SIF is a strong proxy for actual photosynthetic activity in canopies (e.g., Porcar-Castell et al., 2014; Verrelst et al., 2015b, 2016), and over the years various SCOPE-based SIF retrieval





**Table 1.** Overview of widely-used biophysical variables inferable from RTM inversion to assess vegetation productivity information.

| Trait | Description | Key references |
|---|---|---|
| Green leaf area index (GLAI) | GLAI quantifies the photosynthetically active foliage area and is proportional to gross photosynthesis and an important driver of net primary production. | Myneni et al. (2002); Baret et al. (2007, 2013) |
| Fraction of absorbed photosynthetically active radiation (fAPAR) | FAPAR refers to the amount of incoming solar radiation absorbed by live vegetation in the spectral range from 400–700 nm, divided by the total amount of absorbed radiation. | Knyazikhin et al. (1998); Myneni et al. (1997); Gobron et al. (2006) |
| Leaf chlorophyll content (LCC) | LCC refers to the total chlorophyll a+b content per unit leaf area ($\mu g/cm^2$). Chlorophyll molecules are responsible for harvesting the incoming PAR required to drive the light-dependent reactions of photosynthesis. LCC is closely related to leaf photosynthetic capacity. | Croft et al. (2020, 2017); Luo et al. (2019) |
| Canopy chlorophyll content (CCC) | CCC as the product of (G)LAI and LCC quantifies the amount of photosynthetically active radiation absorbed by a canopy and therefore relates to primary productivity. | Ali et al. (2020); Gitelson et al. (2014, 2015) |
| Solar-induced fluorescence (SIF) | SIF is an electromagnetic signal emitted by chlorophyll a of photosynthesizing plants and provides a mechanistic proxy for photosynthesis. | Frankenberg et al. (2011); Guanter et al. (2012); Porcar-Castell et al. (2014) |

schemes have been proposed to derive GPP, usually by taking ecosystem-specific characteristics into account (e.g., Damm et al., 2015; Norton et al., 2019; Pacheco-Labrador et al., 2019; Yang et al., 2022).

### 3.1.3 Gap-filling and smoothing methods

Continuous, complete and unbiased TS data is often a key requisite to monitor vegetation productivity using optical EO sensors. Perturbing factors such as clouds, snow or aerosols overlay the vegetation signal. In addition, insufficient
or irregular frequency of EO data acquisitions causes sparsity of observations in the temporal domain. Both factors may leave large gaps in the TS data stream. Large gaps can have serious implications for monitoring vegetation productivity, leading to biased estimates of variables, loss of information, decreased statistical power, increased





standard errors, and substantial uncertainty in findings (Dong and Peng, 2013). Besides gaps, instrument errors (e.g., due to detector malfunctions and degradation) and directional effects related to the anisotropy of canopies (Forsström

et al., 2021) can bias the obtained vegetation signal and increase the overall uncertainty budget. Notably, the amount of data gaps and noise strongly depends on the season, topography, location or environment (e.g., Beck et al., 2006; Vuolo et al., 2017). Multiple approaches have been presented to obtain continuous TS, from simple temporal compositing approaches to more advanced gap-filling techniques (e.g., Vuolo et al., 2017; Belda et al., 2020a). A high-quality signal can be assumed to represent the true seasonal trajectory of vegetation and should be carefully processed

to retain the short-term character of data (e.g., using smoothing filters or splines). Signals with a high degree of noise need to be constrained by fitting to a predefined function to avoid unrealistic variations (e.g. asymmetric Gaussian or logistic functions). TS filters and splines can to some degree balance between retaining or smoothing short-term variations, and with these methods, parameter settings can be defined that balance smoothness with fidelity to the data (Atzberger and Eilers, 2011b). Important considerations when applying smoothing to TS data are whether data

should be fitted to the upper envelope to compensate for signal bias (Chen et al., 2004; Jönsson and Eklundh, 2004), how to treat data points labelled as sub-optimal quality (e.g. cloud shadow pixels), and how to handle long periods of missing data (Beck et al., 2007; Jönsson et al., 2018; Bolton et al., 2020). Gap-filling and smoothing methods can be categorized into (1) smoothing and interpolation methods, (2) data transformation methods, and (3) fitting methods. An exhaustive overview of available methods is provided in recent reviews by Zeng et al. (2020) and Pipia

et al. (2022b), and is therefore not repeated here. However, of interest in the context of the paper is that gap-filling and smoothing of TS data streams provides improved signal quality for the estimation of environmental variables indicative of vegetation productivity, like carbon fluxes (e.g., Olofsson et al., 2007; Sjöström et al., 2009; Schubert et al., 2010; Fensholt and Proud, 2012; Tang et al., 2013).

## 3.2   Assessment of vegetation productivity using trend analysis and anomaly detection

Long TS data streams of VIs (Sect. 3.1.1) or quantitative traits (Sect. 3.1.2) are particularly well suited for trend analysis, a widely used method for monitoring plant productivity (Eastman et al., 2009). Such analysis includes aspects such as abrupt or gradual changes in trends, as well as timing, number, and direction of such changes (Verbesselt et al., 2010). An example of TS decomposition is shown in Fig. 4. Each of these components can be further analyzed, for example, using separate trend models for annually derived attributes (Stellmes et al., 2013; Munawar and

Udelhoven, 2020).

Regarding trend analysis, the study by Karkauskaite et al. (2017), for instance, explored data from MODIS (from 2000 to 2014) to evaluate the performance of PPI, NDVI and EVI in analyzing the trends of SOS in boreal regions of the Northern Hemisphere. The authors compared the VI trend results with GPP-retrieved SOS derived from a network of flux tower observations. Although all three VIs produced similar trends in SOS, a pronounced land-cover

dependence was observed, with PPI outperforming the other two VIs.

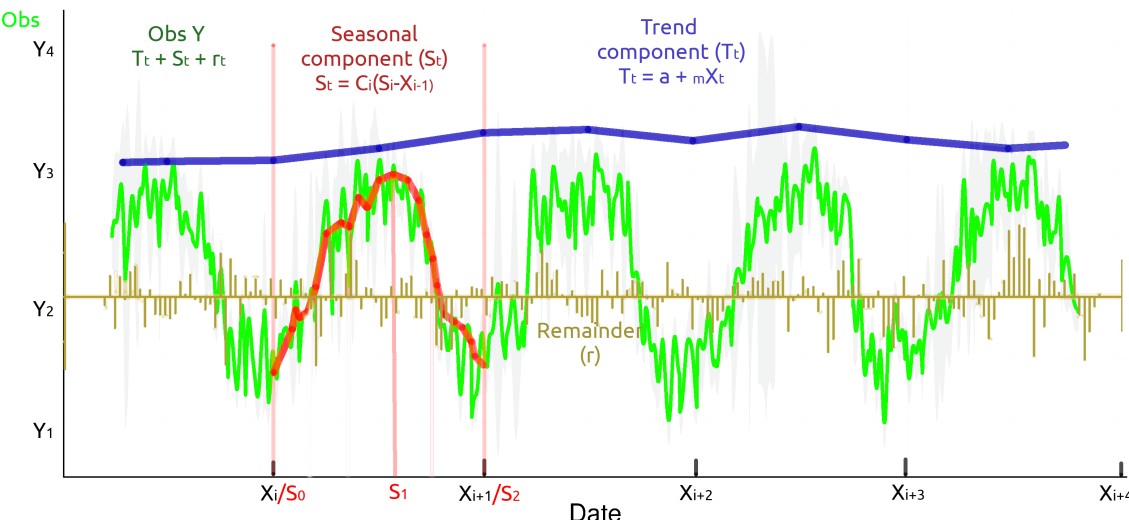

**Figure 4.** Generic plot showing the different components for TS analysis. Point observations result from the coupling of a general trend, a seasonal component, and a remainder quantity. The decomposition allows for measuring the trend at specific points, by disentangling seasonal effects.

In the context of anomaly detection, specific indices have been proposed: the Vegetation Condition Index (VCI) (Kogan, 1995) informs about overall vegetation conditions by referencing actual NDVI values with long-term statistics over the same period. The main application of VCI is related to drought detection (Klisch and Atzberger, 2016; Rembold et al., 2015a). Similarly, the Vegetation Productivity Index (VPI) (Smets et al., 2015) was proposed to detect anomalies in vegetation productivity. Importantly, these methods were developed for natural ecosystems such as boreal forests or sub-tropical savannahs where the vegetation type is assumed not to change from year to year. Thus, these indices are not appropriate for ecosystems with regular changes in species composition; e.g., agricultural croplands with crop rotation schedules. A deep learning approach for forecasting VCI was presented by Lees et al. (2022), demonstrating the usefulness of detecting drought conditions in Kenya using this anomaly index.

In addition to decomposing and analyzing trend patterns of a VI TS, an option is linking the VI to other environmental variables that influence vegetation productivity using distributed lag models (Udelhoven, 2011). However, relationships between climatic variables and responses in VI TS tend to be non-linear, spatially non-stationary and sensitive to the scale of analysis. Simple regression model techniques such as Ordinary Least Squares (OLS) fail





to model vegetation productivity accurately. To overcome such shortcomings, geographically weighted regression
(GWR) approaches were suggested (Georganos et al., 2017).

### 3.3    Assessment of vegetation productivity using land surface phenology

Land surface phenology describes the seasonal timing and duration of vegetative growth using TS of VIs (Sect. 3.1.1)
or biophysical variables (Sect. 3.1.2) (De Beurs and Henebry, 2004). Typical LSP metrics are dates and values for
the start of the season (SoS), end of the season (EoS), length of the growing season (LoS), the peak of the season
(PoS), season amplitude, and steepness of the greening and browning periods (Reed et al., 1994; Beck et al., 2006).
Depending on the vegetation type studied, varying names can be found in the literature, such as the onset of
greenness and the start of senescence for deciduous forests (e.g., Duchemin and Courrier, 1999; Kang et al., 2003;
Badeck et al., 2004). A diversity of mathematical methods have been proposed for extracting the metrics from smooth
seasonal trajectories. Most are based on absolute or relative thresholds of the seasonal amplitude (e.g., Bolton et al.,
2020; Jönsson and Eklundh, 2004), whereas others are purely mathematical parameters, such as inflexion points or
derivatives of different order (e.g., Fisher et al., 2006; Elmore et al., 2012; Melaas et al., 2013). A comprehensive
review of the definition and extraction of LSP metrics is provided by Zeng et al. (2020).

Commonly, LSP metrics are used to study the impact of environmental changes on ecosystems: shifts in LSP, e.g.,
the earlier timing of SoS, indicate climate change (Abbas et al., 2021). For instance, Wood et al. (2021) used three
decades of AVHRR data over the U.S. Northwestern Plains to study the impact of climate change and agricultural
management on LSP. They concluded that phenological indicators might be decoupled from climatic factors due to
anthropogenic interference suggesting that LSP alone could be insufficient to explain changes in productivity.

In general, the concept of LSP has been discussed controversially. Helman (2018) stressed that changes in vegetation
species composition rather than phenological transitions could produce a false-positive signal in LSP, especially in
coarse-resolution satellite data such as AVHRR or MODIS. Moreover, LSP metrics show high sensitivity to the
frequency and temporal coverage of observations as well as cloud contamination (Younes et al., 2021).

### 3.4    Assessment of vegetation productivity using dynamic process models and data assimilation

A more advanced perspective is given by combining remotely sensed data with simulations of plant physiological
processes and its temporal development. Simulated plant growth driven and/or constraint by TS data streams and
environmental covariates can be used to study processes that are not directly quantifiable from the satellite data
itself - such as the amount of AGB increase over time (Delécolle et al., 1992). Here, EO data offer the possibility of
providing a dynamic, spatially-continuous parameterisation of model input variables  (e.g., Bach and Mauser, 2003;
Verhoef and Bach, 2003; Hank et al., 2015).

Process-based dynamic vegetation models can have different levels of complexity concerning their ability to simu-
late biophysical and biochemical processes in plants (e.g., Quillet et al., 2010; Ardö, 2015). Based solely on empirical
data, canopy structure dynamics models (CSDM) have been proposed to simulate a TS of canopy traits such as LAI



as a function of temperature (growing-degree-days) (e.g., Baret et al., 2000; Koetz et al., 2005). Using the concept of LUE, Goudriaan and Monteith (1990) described vegetation dry matter accumulation as a function of leaf area expansion. By including further knowledge about physiological processes and plant morphology, more advanced DVMs
can be created to simulate ecosystem productivity such as for boreal forests (e.g., Liu et al., 1997) or croplands (e.g., Delécolle et al., 1992; Launay and Guerif, 2005; Liu et al., 2016).

Fischer et al. (1997) already distinguished three different strategies to combine remotely sensed TS of vegetation with process-based models, which can be seen as state-of-the-art, as delineated in Fig. 5: (1) model forcing, (2) model recalibration, and (3) coupled forward modelling. In the model forcing strategy (Fig. 5, 1), the remotely observed
state variables (e.g., fAPAR, LAI) are forced (input) into the process model. In the recalibration strategy (Fig. 5, 2), also known as 'data assimilation', remotely sensed state variables are used to readjust DVM parameters or inputs whenever an observation becomes available. While the first two strategies involve inverse modelling to obtain the remotely sensed state variables, the third approach relies entirely on forward modelling (Fig. 5, 3). It couples a DVM with an RTM to simulate vegetation optical properties, which are then compared to remotely sensed data. The
main advantage of this strategy is the avoidance of inverse modelling, which is not only ill-posed but usually also computationally intensive. With this, the method can provide the prerequisite of the 'Digital Twin' (DT) concept allowing to model productivity with high fidelity for longer time periods, and eventually evaluate different future scenarios.

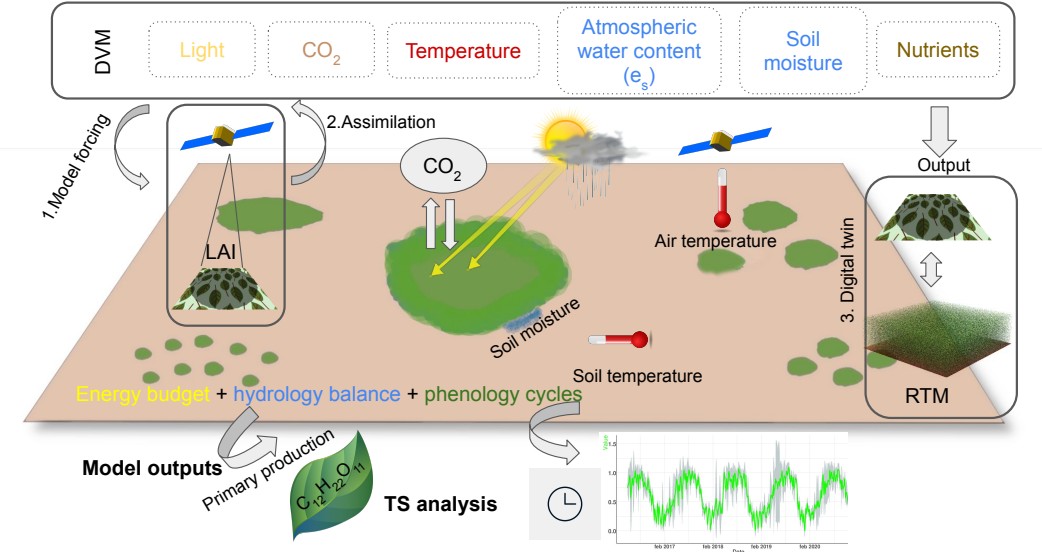

**Figure 5.** Three strategies to combine remotely sensed TS with process-based DVMs: (1) model forcing, (2) data assimilation (including model recalibration and re-initialization) and (3) coupled forward modelling of DVMs and RTMs (Digital Twin).





## 3.5 Toolboxes for vegetation productivity studies

A variety of sophisticated software packages have been developed to facilitate the processing and analysis of large image TS and ultimately provide key information about vegetation dynamics and productivity. In most cases, these packages are openly available, and share common purposes, although they differ in specific features and methodologies. Broadly, we can distinguish toolboxes for TS processing, TS analysis and change detection, traits retrieval, and process modelling (i.e., DVM). Tab. 2 lists the toolboxes according to this categorization, including

functionalities, productivity metrics and implementation. Note that we have compiled this list to the best of our knowledge; however, it is possible that it may not include all existing toolboxes.

TIMESAT (Jönsson and Eklundh, 2004), for instance, is able to transform noisy signals into smooth seasonal curves and to extract seasonality metrics, like SoS, EoS, and LoS, or integrated values. Originally developed for coarse spatial resolution data (e.g. AVHRR or MODIS), with mostly equidistantly spaced temporal observations, recent versions

have adopted the characteristics of satellites with high spatial resolution but infrequent temporal observations, such as Landsat and S2. While TIMESAT uses least-squares methods, the Decomposition and Analysis of Time Series software (DATimeS) (Belda et al., 2020a) expands established TS interpolation methods to over 20 conventional (e.g., Whittaker smoother (Eilers, 2003)) and advanced ML fitting algorithms, like Gaussian process regression (GPR), which is particularly efficient for reconstructing multi-seasonal vegetation patterns (Belda et al., 2020b). In

this way, DATimeS provides interpolated VI/trait values from unevenly spaced TS and associated uncertainties and allows extraction of phenological metrics for each crop and season. DATimeS then also enables the calculation of the same seasonality metrics as TIMESAT, and also has built the option to fuse time series of two data sources, e.g. optical and radar data (Pipia et al., 2019). Apart from TIMESAT and DATimeS, there are other software tools to analyse VI TS data for phenology-related studies including Phenological Parameters Estimation Tool, enhanced

TIMESAT, Phenosat, CropPhenology and QPhenoMetrics (Zeng et al., 2020).

TimeStats (Udelhoven, 2011) goes beyond the extraction of phenological metrics as it expands TS analysis methods to parametric and non-parametric methods for trend detection, generalized-least square regression, distributed lag models, cross spectra analysis, windowed trend and frequency analysis, continuous wavelet transform, and empirical mode decomposition. Based on some of those methods within TimeStats, predefined workflows were implemented

in a web interface called EOTSA (Earth Observation Time Series Analysis) Toolbox (Leopold et al., 2020). EOTSA allows online access to satellite data archives (currently the full PROBA-V database) without the need for local data storage. Fig. 6 shows two examples where NDVI TS were analysed at the continental scale using EOTSA. In the first example, seasonal characteristics (mean NDVI, annual magnitude, peaking time) were derived (step 1), followed by a trend analysis (step 2) (Fig. 6B). The colour composite of the trends for the seasonal characteristics

reveals spatiotemporal patterns. Fig. 6C shows an example of multivariate TS analysis in which NDVI was regressed against lagged rainfall using distributed lag modelling after pre-whitening the TS. Regions with positive correlation





at higher lags depict the dependence of vegetation biomass production on accumulated previous rainfall amounts. These can be for instance located in semi-arid areas occupied by natural grassland.

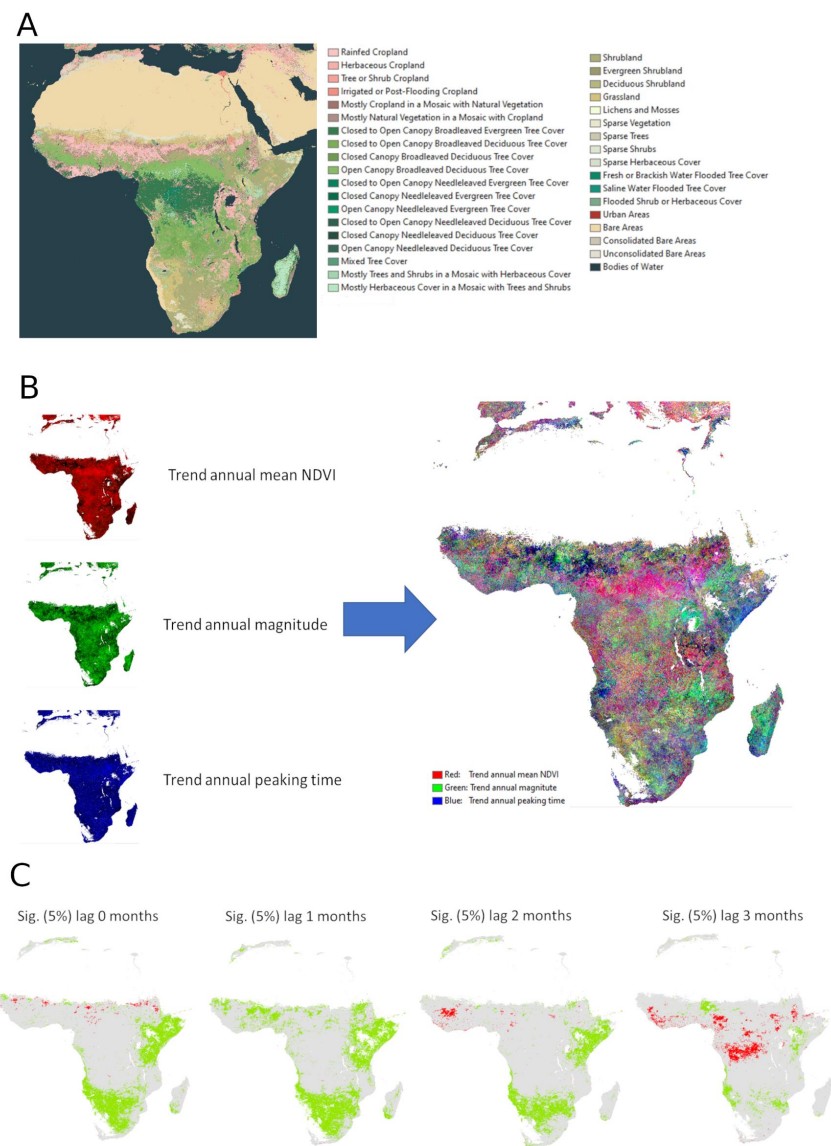

**Figure 6.** Landcover map of Africa and legend (A), trend analysis for a time series of MODIS satellite images (period: 2015-2019) displayed as RGB colour composite (B), results from Distributed Lag Modelling where NDVI was regressed against lagged rainfall (C). Prepared with EOTSA RStudio version.



BFAST (Verbesselt et al., 2010) is a generic change detection approach that considers seasonal, trend and remainder components through iterative estimation of the time and number of abrupt changes within TS, and characterisation of change by its magnitude and direction typically applied in forest monitoring studies.

HiTempo (Van Den Bergh et al., 2012) is a software tool created to aid in the study of TS analysis of hyper-temporal sequences of satellite image data. The platform was specifically designed to simplify the exhaustive evaluation and comparison of algorithms while ensuring the reproducibility of experiments.

SPIRITS (Eerens et al., 2014) is a comprehensive software toolbox designed for environmental monitoring, with a particular emphasis on generating clear and evidence-based information for crop production and decision-makers. SPIRITS provides a vast array of tools for extracting vegetation indicators from image time series and estimating the potential impact of anomalies on crop production (Rembold et al., 2015b). With its user-friendly graphical interface, SPIRITS offers an integrated and adaptable analysis environment that facilitates sequential tasking and provides a high degree of automation for processing chains.

The EnMAP-Box 3 (van der Linden et al., 2015) provides a user-friendly GUI with tools for collecting and visualizing spectral profiles from various sources such as raster images. Furthermore, the QGIS processing framework has been expanded by incorporating many algorithms typically utilized in EO data and imaging spectroscopy analysis for a diversity of ecosystems. The "Agricultural Applications", for instance, provide empirical and physically based trait retrieval strategies which can be explored for deriving productivity information (e.g., Danner et al., 2021).

Regarding RTMs, the Automated Radiative Transfer Models Operator (ARTMO) toolbox (Verrelst et al., 2011) is an outstanding example. ARTMO provides GUI-based access to several leaf and canopy RTMs, and atmospheric RTMs and offers sophisticated strategies for forward and inverse modelling including state-of-the-art ML methods.

Finally, there are a host of available R packages for extracting crop phenology e.g. CropPhenology for extraction of crop phenology from time series based on VIs (Araya et al., 2018), phenofit package - intended for daily vegetation time series and monitoring of vegetation phenology from satellite VIs (Kong et al., 2022), or LPDynR as a tool to calculate the Land Productivity Dynamics indicator (Rotllan-Puig et al., 2021). Moreover, there are Python libraries for phenology and vegetation productivity apps available for ODC. The aim of the ODC initiative is to enhance the worth and influence of worldwide EO satellite data. It does so by offering an open and free-to-use data exploitation structure, and by encouraging a community to cultivate, maintain, and expand the technology and its range of applications (Killough, 2018).

While the aforementioned toolboxes focus on the usage of remotely sensed data only, we found only a few tools that allow users to work with DVMs. Many DVMs are based on FORTRAN programming and lack graphical user interfaces or high-level programming interfaces. The 'Python Crop Simulation Environment (PCSE) has ported old-style DVMs to modern Python programming but, still, considerable coding skills are required to make use of it. PCSE offers a platform for carrying out crop simulation modelling along with tools to read supporting data (such as weather, soil, and agricultural management) and components for simulating various biophysical processes including phenology, respiration, and evapotranspiration. Additionally, PCSE features implementations of widely


used crop and grassland simulation models like WOFOST, LINGRA, and LINTUL3. WOFOST, for instance, has
been employed in the operational crop yield forecasting system MARS, which is used to monitor crops and predict
yields worldwide (De Wit et al., 2019; Lecerf et al., 2019).

**Table 2.** Toolboxes recommended and used for vegetation productivity monitoring using remotely sensed TS. Note that this
list is not necessarily exhaustive, but rather a selection of some of the most notable tools that we are aware of, to the best of
our knowledge.

| Toolbox (link) | Functionality/productivity metrics | Implementation | Reference |
| --- | --- | --- | --- |
| *TS processing:* | | | |
| TIMESAT | Gap-filling / phenology metrics | Standalone | Jönsson and Eklundh (2004) |
| DATimes | Gap-filling / phenology metrics / TS fusion | Matlab GUI | Belda et al. (2020a) |
| *TS analysis and change detection:* | | | |
| TimeStats | Trend, seasonal, and multivariate analysis | IDL virtual machine | Udelhoven (2010) |
| EOTSA | Trend, seasonal, and multivariate analysis | Web interface, RStudio | Leopold et al. (2020) |
| BFAST | Phenology metrics / breakpoint analysis | R Studio | Verbesselt et al. (2010) |
| HiTEmpo | Model-based change detection algorithm | - | Van Den Bergh et al. (2012) |
| SPIRITS | Indicators / anomalies | Java virtual machine | Eerens et al. (2014) |
| R libraries | Indicators, analysis, visualisation | R | e.g., Araya et al. (2018) |
| *Traits retrieval:* | | | |
| EnMAP-Box 3 | Data-agnostic handling of multi-sensor TS data | Python, QGIS plugin | van der Linden et al. (2015) |
| ARTMO | Quantitative traits / RTMs, ML | Matlab GUI | Verrelst et al. (2012) |
| *Process modelling:* | | | |
| PCSE | DVM forward modelling, satellite data assimilation | Python | - |

## 4   Validation of RS-based primary productivity estimates

Validation is a critical step in ensuring the accuracy and reliability of estimated quantities or (vegetation) products
derived from remotely sensed data sets (Justice et al., 2000). The validation process involves comparing the estimates
with those from independent sources, such as *in situ* observations, to evaluate their overall quality and suitability for
a particular application. The comparison between remotely sensed data products and ground-based measurements
enables the detection of errors and biases in the retrieved products and improves the interpretation and understanding
of the underlying ecological processes (Wu et al., 2019). Ultimately, validation plays a crucial role in ensuring the
integrity and usefulness of remotely sensed data.

### 4.1   Validation strategies

In the context of productivity monitoring, we distinguish three distinct validation methods:





1. Manual validation;

2. Local sensor networks;

3. Multi-product intercomparison.

These methods are illustrated in Fig. 7, which showcases how they interplay to provide accuracy, time resolution, and spatial representation, as further elaborated below.

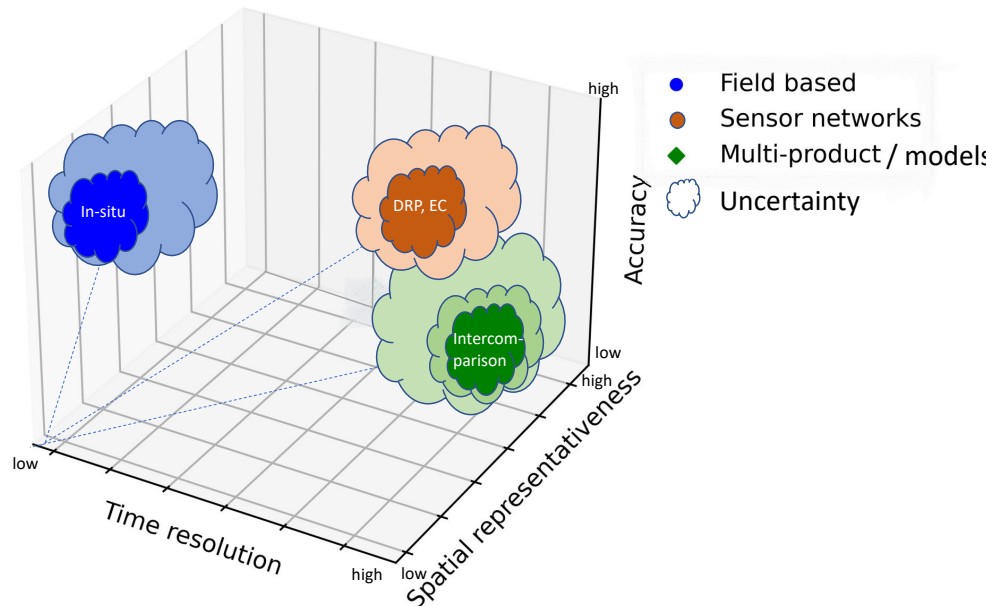

**Figure 7.** Interplay of the three main approaches (field-based: manual validation, sensor networks and multi-products inter-comparison) of validating vegetation productivity, as a function of time, spatial representation and accuracy.

The first category, manual validation (1.), involves comparing RS data products to direct ground-based observations of productivity-related vegetation traits. Examples are the direct determination of GLAI, AGB, and litter biomass. In many cases, *in situ* data is collected by harvesting plots and determining dry biomass (e.g., Zhang and Zhang, 2016; Liu et al., 2021b). One prominent data set is the ORNL DAAC Net Primary Productivity data collection (ORNL DAAC, 2023). It comprises field measurements of AGB and estimated NPP from roughly 100 terrestrial study sites across the globe, including different types of forests, grasslands and crops. These data were gathered from various published literature and other available sources of information. Manual validation provides means of calibrating models to ensure consistency over time, which is essential for long-term studies.





The second category, sensor networks, (2.) is perhaps the most widespread and promising strategy for validating productivity products from EO data. This category refers to a network of distinct sensors, comprising spectral radiometers, phenocams and eddy-covariance (EC) flux towers (e.g., Baldocchi et al., 2001; Baldocchi, 2003; Hilker et al., 2011; Toomey et al., 2015). Such an approach requires deploying validation sites or observation networks with standardized observation protocols (Morisette et al., 2006). The employment of spectroradiometers, phenocams, and

EC systems are valuable tools for providing both continuous, high-resolution (i.e. sub-daily) estimates of vegetation productivity over daily to decadal time-frames, and also serving as validation for satellite-based products. Phenocams capture time-lapse images of vegetation, allowing the monitoring of phenological events such as leaf emergence, flowering, and senescence. This information is valuable for tracking the growth and development of vegetation, as well as for identifying changes in productivity due to environmental stressors. For example, the SpecNet network (SpecNet,

2022) aims to link optical measurements with flux sampling and standardized field optical methods (e.g., Gamon et al., 2006, 2010). The Committee on Earth Observing Satellites (CEOS) group on Calibration and Validation is currently leading efforts concerning the development of best-practice phenology validation protocols and the establishment of ground-reference sites across different biomes (NASA, 2023). The use of EC techniques for providing direct measurements of the exchange of carbon, water and energy between vegetation and the atmosphere (Baldocchi

et al., 2001) has provided an extremely valuable means of measuring plant productivity across diurnal to decadal time scales. The longest-running flux tower is located in Harvard Forest and has been providing continuous measurements at half-hourly intervals since 1989 (Urbanski et al., 2007). Several national and regional networks of flux towers exist (e.g. Ameriflux, Chinaflux, Ozflux, ICOS), which has enabled the contribution of EC data to improve our understanding of plant-environment interactions to go beyond a single site or ecosystem to regional-to-global studies.

To address data consistency and allow cross-site comparisons, FLUXNET was established in 1997, which is a 'network of networks' and has led to harmonised methods and datasets. The latest dataset of FLUXNET, the FLUXNET2015, contains gap-filled time series of GPP, Re and meteorological data for 1500 site years, along with an estimation of uncertainties (Pastorello et al., 2020). However, there are concerns around the spatial and temporal representative of EC data, due to the disproportionate predominance of flux towers being located in North America and Europe (Chu

et al., 2017). To scale from the footprint of individual flux tower sites to gridded, spatially- and temporally-explicit products, a variety of machine-learning techniques have been employed, including neural networks, regression trees and kernel methods (Beer et al., 2010; Jung et al., 2011, 2020). EO data is usually used, along with meteorological data within the ML algorithms, to extrapolate across time and space. These EC-derived products, such as within the FLUXCOM initiative (Jung et al., 2020) have been used extensively in validating other sources of vegetation

productivity estimates, including those from satellite-based remote sensing and also terrestrial biosphere models (Chu et al., 2017).

The integration of these diverse ground-based sensing techniques together with EO data streams is suitable for monitoring large-scale vegetation dynamics, and it can aid in the interpretation and validation of productivity models obtained from remotely sensed data (Balzarolo et al., 2014). From a technical point of view, it is common to find





literature that explores the accuracy of satellite imagery validated through such near-surface sensors. Additionally, there is increasing usage of similar networks focused on different aspects of vegetation and supported by the spread of low-cost and IoT sensors, for example, the TreeTalker network (Valentini et al., 2019; Tomelleri et al., 2022).

The third category, multi-product intercomparison, (3.) involves the benchmarking of multiple productivity products or different models using EO data. This validation approach requires a thorough comparison of the obtained
products with similar ones to check for consistency (Beer et al., 2010; Lin et al., 2022; Meroni et al., 2012). A critical aspect of this approach is ensuring that the models or products being compared are fit for purpose. In other words, they must be appropriate for the specific application or use case. Additionally, the cross-comparison of distinct types of models, such as (an ensemble of different) DVMs and data-driven approaches (e.g., Ardö, 2015; Jung et al., 2020), can provide valuable insights into the strengths and weaknesses of each model type. Benchmarking models using
RS data can help to improve their accuracy and reduce errors in their predictions, which is essential for applications such as monitoring global climate change and assessing the health of ecosystems. It can also aid in developing more advanced primary productivity models that can better account for the complexities of ecological processes and environmental variability.

## 4.2  Bridging the scaling gap

Scaling issues remain one of the most significant challenges in extracting vegetation productivity, regardless of the metric chosen (Zeng et al., 2020; Caparros-Santiago et al., 2021). The disparity in spatial and temporal resolution between *in situ* measurements and remotely sensed data often creates uncertainty in the extracted vegetation productivity estimates. While *in situ* (point) observations are typically species-specific, RS platforms capture a mixture of vegetation types within their large geographic footprint. Consequently, directly comparing *in situ* and remotely
derived productivity estimates can be difficult, if not impossible. Furthermore, while *in situ* observations or local sensor networks provide a high level of detail and accuracy, their geographical coverage is often limited and may not be indicative for large-scale studies (see also Fig. 7). In contrast, EO data products from multiple satellites offer broader coverage, but they suffer from coarser spatial or temporal resolutions. This trade-off between detail and coverage presents a significant challenge in scaling *in situ* observations or local sensor networks (categories 1)
and 2) into the larger scale captured by EO data. Therefore, to overcome the scaling challenge and enhance the accuracy of remotely derived vegetation productivity metrics (see blue box in Sect. 1.1), an effective protocol for the calibration and validation of such metrics using *in situ* observations, sensor networks, and multi-product / model intercomparison is essential. See also the multiscale validation scheme as outlined in Malenovskỳ et al. (2019).

## 5  Systematic literature review on time series based applications for vegetation productivity

This section aims to complement the previous sections by taking a tour across principal thematic applications through a meta-review. We do not assess the calculation of productivity applied in these studies. Instead, we aim to





provide a thorough overview of how remotely sensed TS data were explored to estimate productivity for agricultural, forestry and other natural ecosystem applications. In this way, readers will be redirected towards specific scientific studies analysing productivity with a multitude of proxies and methods for these application domains.

## 5.1 Systematic literature review

The systematic literature review followed the guidelines of the Preferred Reporting Items for Systematic Reviews and Meta-Analyses (PRISMA) (Page et al., 2021). The SCOPUS and Web of Science web catalogues were queried for published, peer-reviewed publications. In SCOPUS, the title, abstract and keywords were searched with the query "time AND series AND productivity AND 'remote sensing' AND (vegetation OR forest OR crop)", while the topic field in the Web of Science catalogue was searched for "'time series' AND (vegetation OR forest OR crop) AND productivity". The resulting 915 records of the two databases were merged into one database by omitting duplicated records as identified by their DOI (Fig. 8). The records were further screened to include research articles and conference contributions in the English language, excluding review studies. Furthermore, the studies were required to use RS analysis of terrestrial vegetation with at least two observations in time. For each entry, a range of attributes was recorded (Tab. 3).

**Table 3.** Attributes retrieved for the systematic review.

| Attributes | Definitions |
|---|---|
| Sensor/platform | Sensors and platforms used in the study |
| Spatial resolution | Ground sampling distance of primary data product used in the analysis (in metre) |
| Land cover | Land cover according to IGBP land cover classes[1], plus category of LULCC |
| Study area category | Size of study area in administrative terms (local, region, country, multi-country, continental, pan-continental, global) |
| Study area size | Size of study area (in km$^2$) |
| Time series start/end [date] | Start and end date of time series |
| Revisit | Frequency of observations in the time series (in days) |
| Time series steps | Number of observations between start and end date (alternative to revisit) |
| Definition productivity | Primary RS products used to derive productivity metrics (VI/LSP/traits/process/LULCC) |

Fig. 9 summarises how studies defined vegetation productivity, and which RS-derived products were used to approximate productivity. Note that some studies referred to productivity but did not specifically state if the generated products were meant to be a representation of productivity, this required care in the interpretation of the results. Generally, productivity proxies were categorized into (1) VIs, being the simple algebraic transformation of spectral observations (see also Sect. 3.1.1); (2) phenological metrics, i.e. derivatives of observations over time, as described in Sect. 3.3), (3) traits, i.e. biophysical/ biochemical properties of vegetation at the time of observation




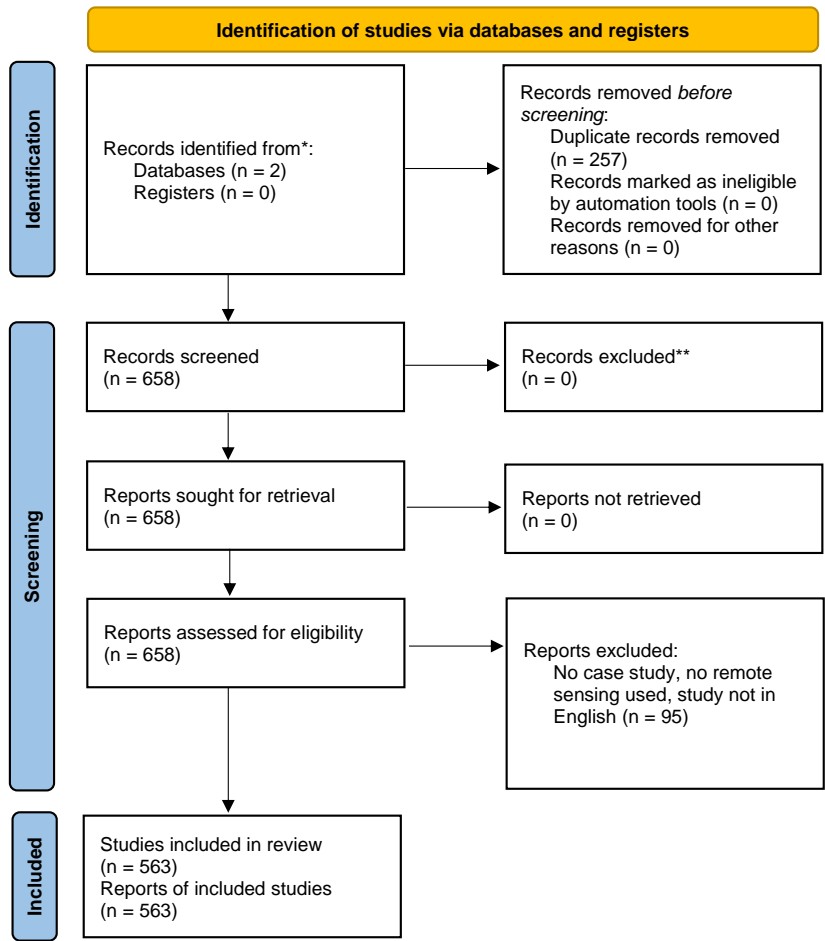

**Figure 8.** Identification of studies via databases and registers according to PRISMA (Page et al., 2021).

(see also Sect. 3.1.2), (4) process, which implies the use of DVM (see also Sect. 3.4), and finally (5) land use and land cover change classifications (LULCC). VIs were most often employed to describe productivity, with almost 50% of all analyzed studies. Specifically, most studies relied on NDVI TS, which may be the most used and well-known method

to analyse TS in the context of vegetation productivity. VIs were followed by traits, processes, and phenological metrics, and cover characteristics as less often used proxies.

Fig. 10 and Fig. 11 show the trends in spatial resolution and sensors underlying the vegetation productivity studies, respectively. In both, three phases can be distinguished: first, the dominance of AVHRR-based and coarse-resolution studies until 2005 (e.g., Wessels et al., 2004). In this period, only 21 studies (3.7 %) were published. Second, a rising


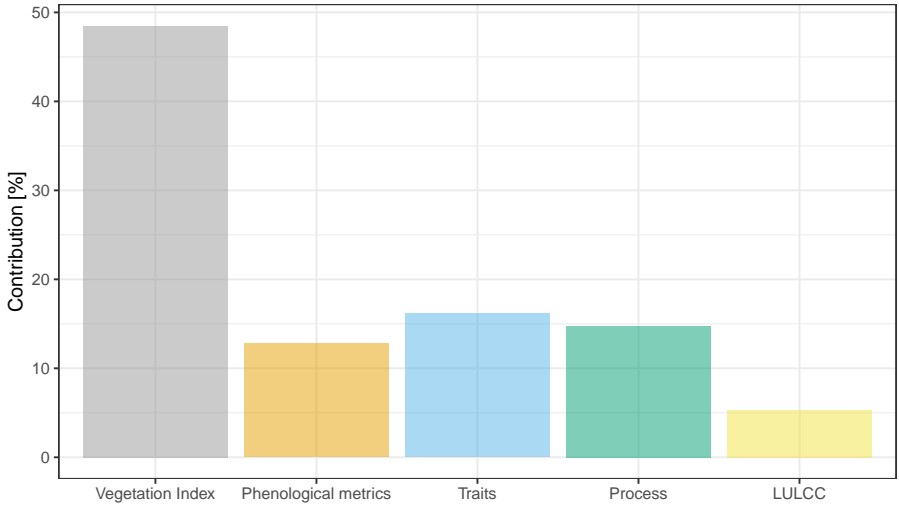

**Figure 9.** Definitions and RS products used in the context of vegetation productivity, i.e. to derive the productivity metrics, in reviewed studies.

contribution of MODIS and Landsat marked the period from 2005 until 2017 (e.g., Boisvenue et al., 2016). During this period also the number of studies per year increased from 6 (1.1 %) in 2005 to 52 (9.2 %) in 2017. In Fig. 10, the years 2013 and 2016 appeared to be outliers with the highest portion of studies with larger than 1000 m resolution. The 2016 outlier can be explained with the publication of the Global Inventory Monitoring and Modeling System (GIMMS) third generation NDVI (NDVI3g) long-term TS dataset based on AVHRR (Pinzon and Tucker, 2014). The last phase started in 2017 and is marked by an increasing trend towards sub-1000 m resolution studies driven by the increased availability of longer-term Landsat and MODIS TS. Studies combining both sensors make up 6.3 % of all (e.g., Knauer et al., 2017; Kussul et al., 2017). Moreover, an unprecedented amount of other sensor TS data became available, see also Fig. 3. Despite the launch of the Sentinel-2A only in 2015, already 29 studies (5.1 %) made use of it for analysis (e.g., Abdi et al., 2021).

In Fig. 12, trends in assessed aggregated land cover types of the reviewed studies are indicated. Category "Other" includes studies covering multiple land cover types as well as land cover/land use change studies. Hereby the dominance of agricultural studies can be clearly seen, followed by multiple, forests (e.g., Boisvenue et al., 2016) and finally grasslands (e.g., Brinkmann et al., 2011).

## 5.2 Agricultural applications

Exploration of TS data has been focused on cultivated areas due to the high significance of agroecosystems for providing global food security. In agricultural applications, grain or fruit yield is often considered the primary metric for productivity. Being indicated as "Vegetation Index" or "Traits" in Fig. 9, these studies used VIs or quantitative

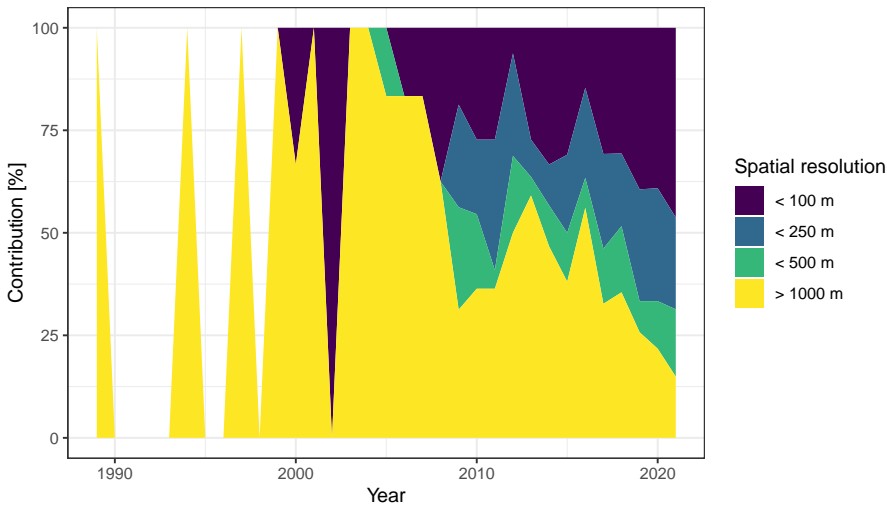

**Figure 10.** Trends in spatial resolutions (pixel size) at which spatial products were produced in reviewed studies.

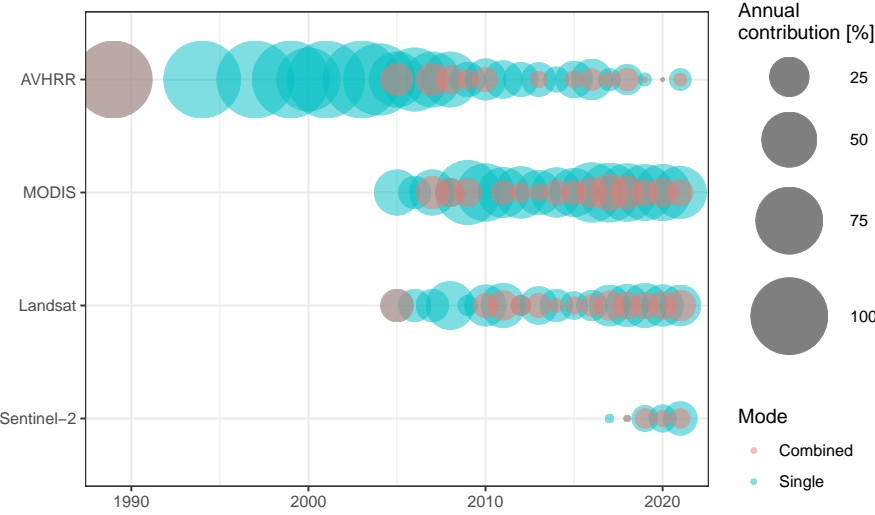

**Figure 11.** Trends of selected sensors/missions in reviewed studies. Combined mode refers to the fusion of the principal sensor with other RS data. Single mode refers to the sole use of the selected sensor for the production of geospatial products.

traits as one of several inputs in data-driven or process models (e.g., He and Mostovoy, 2019; Ma et al., 2021; Guo et al., 2019) or transformed those into phenological metrics, such as calendar / thermal time or LoS (e.g., Duveiller et al., 2013; Azzari et al., 2017) to predict yield. With regards to VIs, mainly NDVI was used to predict crop yield (e.g., Lopresti et al., 2015; Suijker and Medrano, 2018). For instance, corn and soybean yield was estimated

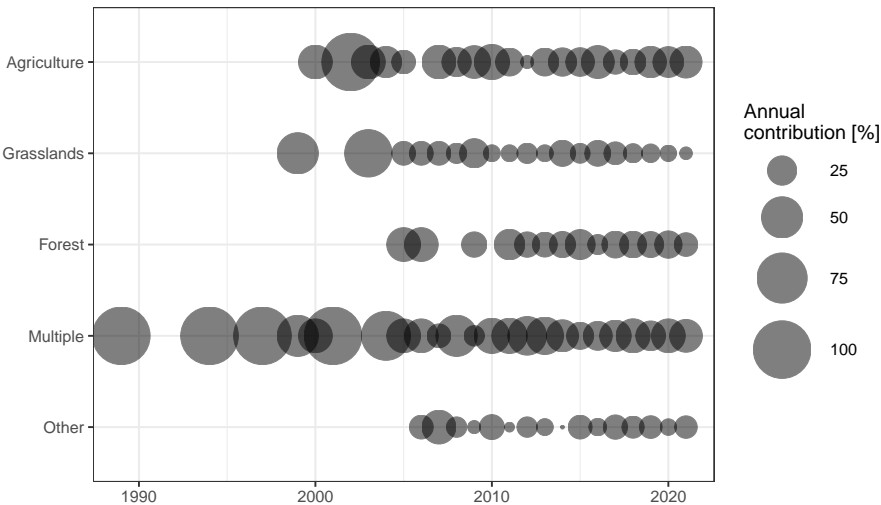

**Figure 12.** Trends in assessed aggregated land cover types in reviewed studies. The category "Other" includes studies covering multiple land cover types as well as land cover/land use change studies.

from six-year TS MODIS-driven NDVI by training regression tree-based models (Johnson, 2014) or for grape the yield was forecasted by training a separate artificial neural network with Landsat NDVI, LAI and normalized difference water index (NDWI) over three years (Arab et al., 2021). In a more complex set-up, Houborg et al. (2015)

retrieved leaf chlorophyll content from Landsat TS data to constrain community land model simulations of GPP; whilst Yan et al. (2009) predicted the seasonal dynamics of GPP using a satellite-based vegetation photosynthesis model (VPM). The inclusion of multiple and heterogeneous data sources as inputs for ML models can improve results for crop yield forecasting. For instance, Perich et al. (2023) evaluated four different methods (including ML and deep learning) for pixel-based, within-field crop yield forecasts for five cereal crops from S2 time-series data

across five years (2017–2021) and 54 fields. While their models showed good performance in general, the results also demonstrated that the ability to predict yield for unseen years varied. This indicates that EO data alone might not be sufficient to explain complex productivity metrics such as yield. The importance of climate TS data, such as maximum temperatures and accumulated rainfall, along with EO data when training ML models was emphasised for crop yield forecasting by Kamir et al. (2020).

As a more direct proxy of plant photosynthetic activity, SIF may be able to directly indicate yield or agricultural production. The study by Somkuti et al. (2020), for instance, showed the potential of integrated GOSAT-derived SIF TS data to estimate crop yield. This research line of yield prediction has since then been adapted using other satellite sources of SIF (e.g. GOME-2, TROPOMI), thereby confirming that SIF contributes to improved yield prediction models (e.g., Peng et al., 2020; Sloat et al., 2021; Li et al., 2022).





## 5.3 Applications in forestry

In the last three decades, the bulk of research on forest productivity TS has focused on estimating AGB, and thus carbon sequestration, and better understanding the role of forests in regulating the climate. To accomplish these tasks, various types of data have often been combined, such as TS of satellite multispectral data (Landsat) with Light Detection and Ranging (LiDAR) and radar (e.g., Powell et al., 2010; Pflugmacher et al., 2014; Nguyen et al., 2020). Forest disturbances can play a crucial role in ecosystem dynamics affecting productivity. Therefore, TS analysis of forest productivity is a fundamental tool for analyzing the magnitude and frequency of such events. Many of these forest disturbances are related and have increased due to climate change. Storms and forest fires have been highlighted as the most significant abiotic disturbances in Europe in recent years (Senf and Seidl, 2021). In addition, satellite TS analysis shows an increased trend in the frequency and intensity of droughts, and illustrative studies have been conducted for northern Europe (e.g., Reinermann et al., 2019; Senf et al., 2020; Descals et al., 2023). Climate change is also increasing the frequency of biotic disturbances like insect outbreaks (Senf et al., 2017; Olsson et al., 2017). All such factors put our forests under increasing pressure and limit the forests' role as the global carbon sink. Therefore, it is essential to monitor temporal and spatial patterns of forest productivity by adopting suitable tools like EO data. To this end, many novel initiatives are related to this specific ecosystem. Examples are actions based on available data like the FAO's Global Forest Observations Initiative (GFOI) (Penman et al., 2016). In recent years, the main concern about forest science has moved towards accurate carbon stock and fluxes estimations, for which the focus on EO has shifted from multispectral proxy productivity estimations to precise forest extension and AGB measurements, by means of LiDAR, radar and SIF instruments. Remotely sensed SIF has become a popular method to study temporal variations of deciduous and evergreen forests. In particular, SIF retrieved from GOSAT (Lee et al., 2013), GOME-2 (e.g., Koren et al., 2018; Getachew Mengistu et al., 2021) and TROPOMI (e.g., Doughty et al., 2019) were used to study the relatively subtle seasonal variations in tropical forests and provided new insights on vegetation activity during the transitions between wet and dry seasons.

## 5.4 Applications for natural ecosystems

Among natural ecosystems, several studies focused on African semi-arid ecosystems (Sahel, South Africa) (Fensholt et al., 2013), the Arctic tundra (Beamish et al., 2020), the northern taiga (Canada, Alaska, Siberia and Scandinavia) (Fiore et al., 2020) and the Middle-Asian grasslands, specifically in the Tibetan Plateau (You et al., 2019; Liu et al., 2020) and the Chinese-Mongolian area (Tüshaus et al., 2014; Gao et al., 2017). Regarding productivity, these studies assessed droughts, fires, changes in phenology metrics, land degradation and vegetation mortality (e.g., Mayr et al., 2018; Buitink et al., 2020). The most used EO missions are MODIS and AVHRR, mainly through the analysis of NDVI and GPP TS, or to a lesser extent, other proxies (EVI, fAPAR) or metrics (NPP) of productivity (e.g., Rankine et al., 2017; Lara et al., 2018). The spatial coverage of these studies is global or regional, while the temporal extent is decadal, as MODIS and AVHRR cover a larger time period, from the 1980s to the present.





Multiple studies took advantage of MODIS ready-to-use products (NDVI, EVI, fAPAR, GPP or NPP), which are compatible with specific phenology analysis software, see also Section 3.5. The MODIS GPP algorithms were also used in GPP estimation studies (e.g., Feagin et al., 2020) and for a comparison with the LUE-based VPM (Liu et al., 2011). Studies also compared the results of estimating GPP using MODIS and S2 data (e.g., Cai et al., 2021).

More recently, satellite SIF data became a valuable source for productivity estimations in natural ecosystems, among others to better capture seasonal periods of water stress and early-season GPP dynamics in drylands (Smith et al., 2018; Wang et al., 2019). Similarly, Merrick et al. (2019) studied satellite SIF data for different biomes, such as grasslands and savannas (among others). The authors concluded that the inclusion of SIF facilitated the differentiation of various vegetation types based on their functional characteristics and seasonal changes, explaining differences in year-round productivity dynamics. Despite being one of the most sensitive ecosystems, wetlands have been the least researched in the TS context. This might be related to their complexity. Traditionally, wetlands were studied using MODIS TS and EC flux tower data (e.g., Kang et al., 2018). Analyses for wetlands are also oriented towards classifying changes in wetland extent on multi-temporal S2 imagery. Products derived from multi-temporal data were used (S2, Landsat), like NDVI, for instance to model NPP by means of the Carnegie-Ames-Stanford Approach (Zhang, 2021; Zhang et al., 2022a).

## 5.5 The role of productivity as a sink for carbon across ecosystems

Vegetation productivity, or GPP specifically, characterizes the "gross" terrestrial carbon sink, the gross amount of $CO_2$ annually sequestered by vegetation. NPP corresponds to the net carbon gain by plants, as it is the difference between the carbon produced by GPP and Ra (Fig. 13). The appropriation of NPP is also a measure of vegetation contribution to climate change mitigation (Alexandrov and Matsunaga, 2008). Overall, knowledge about the productivity of aboveground carbon stocks in forests, agriculture, and natural ecosystems is essential for global climate scenarios (Erasmi et al., 2021). Terrestrial ecosystems, along with the oceans, serve as a natural buffer that restricts the increase of $CO_2$ in the atmosphere by absorbing and sequestering nearly half of emitted $CO_2$ (Friedlingstein et al., 2022).

Fig. 13 delineates the different levels of productivity which are GPP, NPP, NEP and Net Biome Productivity (NBP) with respect to their carbon loss processes and flux densities over time. Distinct ecosystems vary a lot in this respect. While an agricultural system (non-permanent crops) is releasing usually all carbon during one season, natural and forest ecosystems can store carbon for decades, depending on climate, but also on the amount of woody vegetation and tree age (Machwitz et al., 2015). This emphasizes the crucial role of TS data acquisition in monitoring the development of these carbon pools.

The largest terrestrial carbon sink is the world's forests. In a general view, the global forest carbon stock in 2020 was 662 Gt (FAO, 2020), from which 44% is contained in the AGB. The tropics have the largest proportion of the world's forests, and hence they are highly relevant in terms of global climate regulation. However, tropical rainforests are under threat due to deforestation, logging, or cultivation, among others. These human activities lead to the loss





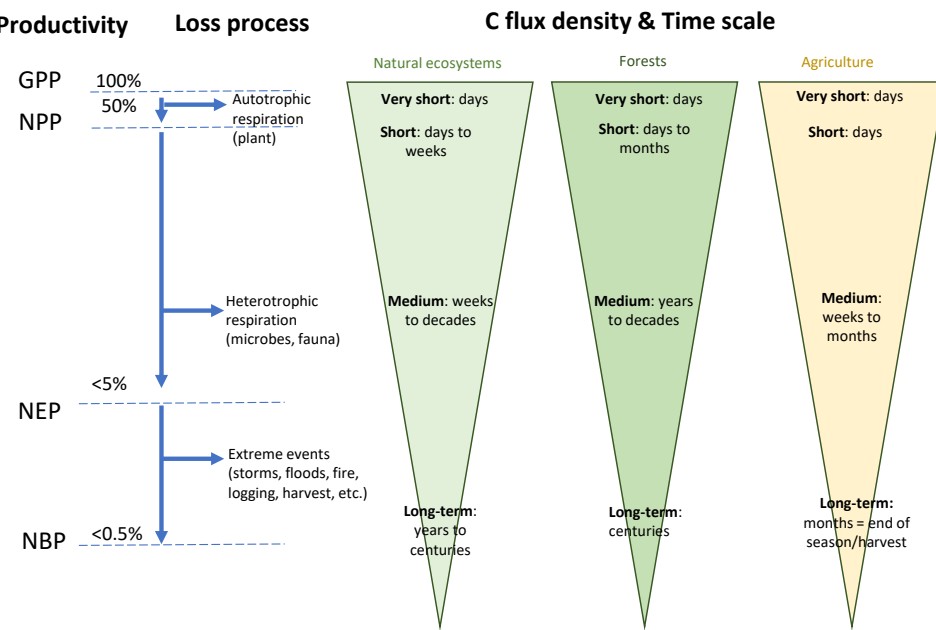

**Figure 13.** Overview of productivity terms along with their loss processes and C flux density over time scales and for the main ecosystems. Note that the given percentages are common average values and may vary.

of biodiversity and carbon storage, and thus to a transition of being carbon sources for the atmosphere. Also, other forest types play essential roles in this context, such as needle-leaf forest systems. A study in southern Sweden, for instance, focused on the plantation of needle-leaf trees (Grelle et al., 2023). The stands underwent a transition from

positive (sources) to negative (sinks) annual carbon fluxes approximately 8 to 13 years after disturbance, influenced by site productivity and management, with net carbon gains of around 5 tC ha$^{-1}$ year$^{-1}$. Additionally, tree crop-based agroecosystems, such as vineyards, have been linked to carbon storage facilities, with fixation rates up to 7.23 tC ha$^{-1}$ year$^{-1}$, where the contribution of root systems implies 9-26% (Brunori et al., 2016) (which, however, cannot be quantified by EO techniques). A recent study on carbon densities simulation in woody vegetation highlighted the

need to improve the integration of models and TS data for a better understanding of the global carbon cycle (Bultan et al., 2022).

     Due to the vast expansion of cultivated surfaces worldwide, the role of productivity of managed land within the global carbon cycle has also increased significantly and deserves particular dedication. Hereby, different factors have been found to influence the dynamics of carbon as sources or sinks, such as climate, tillage measures, fertilization

or irrigation, among others (Luo et al., 2010). For example, tillage usually leads to a loss of soil organic carbon by





organic decomposition, however, in combination and depending on other management practices a larger amount can be stored again by the crops (Haddaway et al., 2017).

# 6 Challenges and outlook

Our review revealed that multiple challenges exist to accurately estimate vegetation productivity from remotely
sensed time series data. In this final section, we will discuss the main challenges, and priority areas of research giving an outlook towards required efforts.

## 6.1 Key challenges

**Efficient use of increasingly available and longer time-series datasets:** Over the past few decades, there has been an increasing number of TS datasets made available, with enhanced spatial details as depicted in Fig. 3.
Despite this progress, there may still be limitations in terms of the spatial and temporal resolutions for specific objectives and applications. Some of these limitations include gaps in available long-term datasets due to persistent cloud cover, and discontinuity of sensors, leading to uncertainties in variable retrievals being relevant for vegetation productivity metrics. While with the advent of cloud-computing platforms access to EO data TS has never been as easy as nowadays, access to other RS resources (local airborne campaigns) is more fragmented and not always open.
From a user's perspective, currently, there is a broad range of high-level, free and easy-to-use toolboxes available (as shown in Tab. 2), allowing and facilitating efficient processing.

**Processes and factors affecting vegetation productivity:** One important goal will be to adopt TS datasets to develop and test methods for heterogeneous natural environments, as the current focus is largely on croplands (see Fig. 12). Satellite TS data streams can effectively capture, characterize, and quantify the spatiotemporal variation
of natural processes. However, some approaches (discussed in Sect. 3) may only provide a relative characterization of vegetation productivity. To increase our understanding, process-based models (i.e., DVMs) are required to provide the mechanistic basis (linking to modelling domains) necessary to capture productivity variations in natural environments. Integrated modelling is a potential approach to address this need.

**Availability of validation data and approaches:** Effective validation of remotely sensed products and derived
productivity metrics is essential for ensuring their reliability and usefulness for a variety of applications. However, the collection of appropriate validation data (see Sect. 4), can be challenging for certain ecosystems. Moreover, there may be discrepancies in the terminology used by different scientific communities, which can hinder effective communication and collaboration. Thus, it is crucial to facilitate interdisciplinary exchanges to promote a common understanding of the terminology and concepts related to RS, as well as to foster collaborations that can help address
knowledge gaps and advance the field.

**Use of TS-based approaches to identify drivers of productivity change:** TS-based approaches can be utilized to identify the drivers of productivity change, particularly concerning critical societal issues such as de-



forestation, land degradation and climate change. These approaches can help to understand better the underlying causes and mechanisms of such changes, which is essential for developing effective strategies to address these issues.

By providing a long-term perspective and enabling the detection of subtle changes over time, TS data can also support more accurate predictions of future trends and impacts. Therefore, the use of TS-based approaches can play an important role in informing decision-making processes and promoting sustainable development.

**Role for deep learning (DL) approaches to be adopted for explorative analysis and support system understanding:** Classical RS data analysis methods for vegetation monitoring (including productivity studies)

usually require (manual) selection of appropriate features from the input data, for instance, spectral indices, texture metrics or temporal segments. The abundance of ways to derive such variables makes it difficult to find the most effective set of predictors for the automated identification of disturbances in vegetation productivity. Deep learning (DL) has been identified as a powerful method that can learn the most appropriate data transformations to get the most relevant data features for solving a specific problem (Kattenborn et al., 2021; Cherif et al., 2023). However, a

key condition is the availability of a large training dataset, i.e., if the dataset is small, then conventional ML methods may be more suitable. For analysing data streams of temporal dynamics, recurrent neural networks (RNN) are of particular interest because they can recognize temporal patterns regardless of data gaps due to missing images or cloud cover. Similar to convolutional neural networks (CNN) for spatial patterns, RNNs make a selection of temporal features (e.g., trend, phenological indicator) obsolete (Kattenborn et al., 2021). A combination of RNN and CNN

potentially enables an end-to-end processing scheme in the spatial and temporal domain and is considered by some authors a potential game changer in analysing TS data for vegetation applications such as productivity studies (Reichstein et al., 2019). Transformers are an alternative to RNNs that originate from natural language processing. Like RNNs, transformers are designed to process sequential input data, but unlike RNNs, they process the entire input all at once, which allows for more parallelisation and reduced training times.Very recently, transformers have

started to advance into RS applications (Aleissaee et al., 2023).

### 6.2 Development of an integrated modelling approach towards the Digital Twin (DT) concept

We suggest that productivity research should focus on the integration of suitable multi-domain radiative transfer models (e.g. SCOPE) with process models (i.e., DVMs) (Moulin et al., 1998; Delécolle et al., 1992) to build "Digital Twins" (DTs) of various ecosystems (Berger et al., 2022). This concept of real-time virtual representations allows

us to mirror behaviour and states over the lifetimes of ecosystems and thus has the potential to overcome current limitations. Therefore, we should aim to develop a conceptual DT framework which implies a DVM with a fully integrated RTM for efficient vegetation productivity monitoring using RS time series. Such integrated models directly simulate remotely observed signals based on the status of the underlying DVM at a point in time. This means the coupled DVM-RTM simulates the spectral signatures of the canopy (400 nm to 2500 nm) as well as SIF emission,

along with physiological processes. To effectively assimilate the sensor data, it is necessary to create strategies that consider the varying availability of data from different sensors and sensor modalities over time. These strategies



should enable continuous updates to the model, allowing for partial assimilation of variables while maintaining internal constraints and connections to additional variables.

Utilizing DVMs, which offer a continuous sequence of temporal growth dynamics, can be beneficial in identifying anomalies and enhancing the creation of continuous system descriptions. In this regard, incorporating a comprehensive numerical model, which assimilates such data, represents a significant advancement. Moreover, to accelerate the forward simulations, surrogate models or emulators can be used to replace some of the more intricate models (Verrelst et al., 2019b). Since DVMs are partly driven by weather variables, they may also allow the integration of weather forecasts and/or climate scenarios in such model designs. This dynamic nature would meet the requirements of a DT of ecosystems, which besides the representation of the current status, also predicts their future behaviour (Verdouw et al., 2021).

## 7 Conclusions

Monitoring vegetation productivity is critical for understanding the health and functioning across ecosystems. In recent years, the increasing availability and quality of optical TS data streams resulted in a large-scale use for monitoring vegetation productivity metrics (e.g. GPP, NPP, yield) for a range of application domains adopting both RS-derived phenological indicators and increasingly more complex integrated modelling approaches. In this review, we identified a vast number of studies that used remotely sensed TS data streams and distinct methods for inferring productivity metrics. These efforts led to valuable insights into vegetation dynamics across ecosystems, including agriculture, forests, grasslands, and others. As the perhaps most urgent topic nowadays, spatially-explicit estimating vegetation productivity is crucial for understanding the role of ecosystems in the carbon cycle. By using satellite data to estimate the amount of carbon stored in vegetation, we can better understand the impact of land use changes (e.g., deforestation) and other human activities on the global carbon balance, and thus climate change. The availability of long-term satellite TS datasets with improved spatial and temporal detail has increased steadily over the past decades. More recently, the emergence of routinely-acquired SIF products proved to provide a more direct linkage towards photosynthetic activity and became increasingly integrated into vegetation productivity processing chains. Validation efforts in ensuring the accuracy, robustness, and reliability of RS-based productivity estimates are another essential aspect of the processing chains. Validation can be performed in several ways, including *in situ* measurements, local sensor networks, and inter-comparisons of available productivity products or models. The definition of harmonized validation strategies is critical to ensure that the methods used to infer productivity from EO data are robust and accurate across various ecosystems and conditions. It also provides confidence in the data and models used to inform management decisions and climate change mitigation strategies. Additionally, we foresee that due to the advancements in ML/ AI, the processing approaches of TS data streams will diversify, and at the same time, the modelling approaches will significantly advance towards holistic processing and representations. Our proposed conceptual framework of a Digital Twin aims to address the limitations of existing approaches and may





provide more accurate and efficient productivity estimation supporting the management of ecosystems at varying scales.


*Author contributions.* Conceptualisation: LK, KB, MM, MS, JV, HA; Formal analysis: BB, DG, ET, FF, GK, IH, LK, LG,
MK MR, MM, OR, RD, SRK, VEG; Investigation: KB, JV, ET, MR, MM; Methodology: LK, KB, HA, JLR, LVG, BB,
JV; Software: KB, EP, JV; Supervision: LK, KB, JV, HA, JLR; Visualization: DG, MM, PRM, MS, KB, AH, BB; Writing -
original draft: LK, KB, JV, HA, JLR, MM, LVG, BB, CA, VGM, EP, GK, BB, ET, OR, MS, SB, CSF, SRK, PRM; Writing
- review and editing: LK, KB, MM, MS, JV, HA, JLR, ZC, LVG, MR, MK, SC, GK, VGM, JP, HC, ETG, MK.

*Competing interests.*

Herewith we declare that no competing interests are present.



*Acknowledgements.* The research was mainly supported by the Action CA17134 SENSECO (Optical synergies for spatiotemporal sensing of scalable ecophysiological traits) funded by COST (European Cooperation in Science and Technology, www.cost.eu (accessed on 16/05/2023)).



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
