# Peer review of "Reviews and syntheses: Remotely sensed optical time series for monitoring vegetation productivity"

_Biogeosciences, 2023_

## Referee Comment (RC1)

Review of *Remotely sensed optical time series for monitoring vegetation productivity.*

Overall, I think the authors did a good job at reviewing the state of the literature surrounding using EO for time-series analyses of productivity. While it was quite long, even for a review paper, it read well and contained a lot of interesting information.

I do feel as if there could be some minor improvements made to various aspects of the paper regarding the following sections:

- The systematic literature review is a bit lengthy while adding relatively little to the overall paper. I believe it could be retained but shortened. For instance, the figures could be compressed into one, multi-panel figure. Figure 8 could be removed. And overall writing could be more efficient.
- Some advances have been made in linking RTMs with DVMs and at least some of them should be cited. A few that I know of:
    - Shiklomanov et al., 2021.
    - Wang et al., 2021, Braghiere et al., 2023 (and other papers regarding the CLIMA Land model).
    - Poulter et al., 2023.
        - Full discloser, I am a coauthor on one of these publications, but I leave it up to the authors to discern and cite the most relevant ones.
- There are a number of current and future missions missing from figure 3, such as EnMAP, EMIT, PACE, and SBG (to name a few).
- "NBP" is missing from the productivity definition box but is discussed later on. Along these lines, figure 13 doesn't quite make sense to me. Is this saying that NBP is 0.5% of GPP?
- Figure 7 could be improved.

---

## Author Comment (AC2)

Dear reviewer,

We would like to thank you for your critical feedback on our paper. We appreciate your time and effort, and we are committed to addressing your concerns.

We agree that the manuscript is lengthy and that more detail is needed in some parts. Mainly, this concerns the introduction, which should more clearly outline the need for a new review and formulate a clear research question.
We reformulated our objective, aligned with the advice of the reviewer, into the following research question: "**What are the state-of-the-art methods for estimating vegetation productivity using remotely sensed time series data, and what are the key gaps, challenges and opportunities for further improvement?**"
This question will be included in the revised manuscript and clearly positioned in the introduction section as soon as the revision is elaborated. As indicated by the reviewer , in Section 3, we will provide a clear overview of the methodologies used to derive the productivity metrics that we reviewed for this manuscript.

Our main focus lies on the precise remote sensing-based estimation of productivity with consideration of the trend toward the increasing availability of higher spatial resolution EO data. Global change is resulting in a landscape, which is more fragmented, scattered and characterized by small patterns. One example is the upcoming trend of agroforestry to make agriculture more resilient. As a consequence, the analysis of productivity needs to integrate high spatial resolution remote sensing data and we preferred to focus more on the spatial scale than on the minimum number of time steps. One of the main keywords of our systematic literature review was "time series." We did not initially define a minimum number of consecutive observations for inclusion in the review. Unlike other papers, which define a time series as consisting of a minimum of several observations, we included studies with a minimum of two images without an upper limit. This allowed us to include studies that have traditionally been labeled under the topic of change detection analysis.

We chose to do this for two reasons. First, we believe that the minimum number of observations in a time series is arbitrary, and we wanted to take a more comprehensive approach to examining the aspect of time. Second, the number of studies using long time series consisting of tens to hundreds of high-resolution (10-30 m pixel size) images is relatively small. If we had only looked at long time series, we would have excluded many studies that observe productivity from Landsat and Sentinel-2 satellites.

The following figures show the number of published papers per number of observations in a time series. Two observations emerge from these figures: i) A relatively large number (about 37) of studies mention the term time series but are based on only 2 images; ii) at a larger

number of observations (n) there is a normal distribution going up to 1000 observations with a slowly decreasing number of papers (p) with increasing n.

Based on the suggestions of the reviewer, we have analyzed this further, and we will add this aspect (including the figure) to the description in section 5.1.

Regarding your point: "...while multiple sections discuss VIs and RTMs as proxies for productivity, my expectation was that the authors would concentrate on reviewing the methodologies to derive the productivity metrics they defined in table 1 using VIs etc…"

Please note that VIs or RTMs are needed to derive information about productivity metrics as listed in the blue box. It is therefore crucial to concentrate on these methods, which are not always proxies. VIs have been used often as proxies, but RTMs are not direct proxies; they provide traits that can act as proxies or can be further used in process models to derive productivity metrics (like VIs).

Please note that **Section 1** is the introduction section. In the introduction section, we want to give an overview of the different concepts, sensors, and methods, which are repeated but explained more in detail in the following sections, as also referred to. However, since it appears to you that we repeat the same information, we will make sure to delete redundant sentences.

Specifically, we will prioritize refining **Section 1.1** to ensure that it provides a comprehensive and accurate overview of photosynthesis, and we will review the literature to add any relevant references that we may have overlooked.

We will also strongly revise **Section 1.2** to ensure that it is organized and that information is not repeated in other sections. In **Section 1.3,** we will emphasize that we estimate productivity using remote sensing data rather than measuring it directly. We will also clarify the relationship between VIs and other variables and productivity metrics.

Furthermore, we will revise **Section 2.4** to ensure that it is more focused on productivity metrics.

**Sections 3.1.3 to 4.2**: We will revise these sections to clarify and state their main objectives and focus them more directly on the productivity metrics that we are discussing. We will also ensure that these sections are consistent with the overall objectives of the manuscript.

**Section 5.1:** We will link this section to GPP and AGB more explicitly, as we have defined these as productivity metrics. We will also discuss the relationship between VIs and GPP/AGB in more detail.

We will also revise the application section as best as possible to make it more focused.

We would like to thank the reviewer again for their thoughtful and constructive feedback. We are particularly grateful for your insights into the structure and organization of the manuscript, as well as your suggestions for how to improve the clarity and focus of our writing. We are committed to addressing your concerns to the best of our ability. Therefore, we will carefully consider all your comments and make appropriate revisions to the manuscript. We are confident that we can produce a revised manuscript that meets the reviewer's expectations and makes a significant contribution to the field.

[Figure]

(33 studies omitted with >1000 observations)

[Figure]

(321 studies omitted with >100 observations)

**Response letter reviewer 1:**

| Reviewers suggestions | Our response | Changes in the manuscript |
|---|---|---|
| Overall, I think the authors did a good job at reviewing the state of the literature surrounding using EO for time-series analyses of productivity. While it was quite long, even for a review paper, it read well and contained a lot of interesting information.
I do feel as if there could be some minor improvements made to various aspects of the paper regarding the following sections: | Thanks for your positive reply… | |
| The systematic literature review is a bit lengthy while adding relatively little to the overall paper. I believe it could be retained but shortened. For instance, the figures could be compressed into one, multi-panel figure. Figure 8 could be removed. And overall writing could be more efficient. | | |
| • Some advances have been made in linking RTMs with DVMs and at least some of them should be cited. A few that I know of:
o Shiklomanov et al., 2021.
o Wang et al., 2021, Braghiere et al., 2023 (and other papers regarding the CLIMA Land model).
o Poulter et al., 2023.
Full discloser, I am a coauthor on one of these publications, but I leave it up | | |

| | | |
|---|---|---|
| to the authors to discern and cite the most relevant ones. | | |
| • There are a number of current and future missions missing from figure 3, such as EnMAP, EMIT, PACE, and SBG (to name a few). | | |
| • "NBP" is missing from the productivity definition box but is discussed later on. Along these lines, figure 13 doesn't quite make sense to me. Is this saying that NBP is 0.5% of GPP? | | |
| • Figure 7 could be improved. | | |
| | | |
| | | |
| | | |

---

## Author Response (AR1)

**bg-2023-88:**     **point-by-point reply to the comments**

Dear Editor, dear Reviewers,

Thank you very much for considering our manuscript, entitled: "Reviews and syntheses: Remotely sensed optical time series for monitoring vegetation productivity," submitted to Biogeosciences Journal. We appreciate your positive feedback and will provide a revision of our manuscript considering the suggestions of the two reviewers.

In the following, we have provided answers to each of your questions and indicated all changes made to the revised manuscript. Please note that all changes in the revised manuscript file are indicated in red.

**Reviewer 1:**

| Reviewer suggestions | Our response | Changes in the manuscript |
|---|---|---|
| Overall, I think the authors did a good job at reviewing the state of the literature surrounding using EO for time-series analyses of productivity. While it was quite long, even for a review paper, it read well and contained a lot of interesting information.
I do feel as if there could be some minor improvements made to various aspects of the paper regarding the following sections: | Many thanks for your positive reply, good suggestions, and the time you dedicated to reading our manuscript. We agree with your points and will provide a revised manuscript; see our responses below. | |
| The systematic literature review is a bit lengthy while adding relatively little to the overall paper. I believe it could be retained but shortened. For instance, the figures could be compressed into one, multi-panel figure. Figure 8 could be removed. And overall writing could be more efficient. | Thanks for your good suggestions! We decided to move Figure 8 to the appendix.
We also provided a multi-panel figure, joining the former Fig. 9-12.

Overall writing has been improved by writing more efficiently and more focused on the productivity metrics. See the changes in the manuscript in red or some text examples in response to reviewer 2. | **Section 5, new figure 7:** |

| | | |
|---|---|---|
| | Please also note that we included a new figure (Fig. 8) indicating the number of observations used by the studies. |
[Figure]
 |
| • Some advances have been made in linking RTMs with DVMs and at least some of them should be cited. A few that I know of:
o Shiklomanov et al., 2021.
o Wang et al., 2021, Braghiere et al., 2023 (and other papers regarding the CLIMA Land model).
o Poulter et al., 2023.
Full discloser, I am a coauthor on one of these publications, but I leave it up to the authors to discern and cite the most relevant ones. | Thanks a lot, these references are indeed relevant and highly interesting; therefore, they were included in the respective sections with corresponding explanations: | **Section 3.4:**
*Shiklomanov et al., (2021), for instance, coupled three existing models, namely the Ecosystem Demography model version 2 (ED2, (Medvigy et al. (2009) PROSPECT-5 (Feret et al., 2008) and a simple soil reflectance model to the EDR model. Their model predicts the full range of high-spectral-resolution surface reflectance, which is dependent on the current state of the ED2 model. Another relevant example is provided by (Wang et al., 2023) with Climate Modeling Alliance (CliMA) Land, which is able to simulate productivity metrics such as GPP, transpiration, as well as canopy reflectance and fluorescence spectra that can be observed by satellites in a high temporal resolution. The authors demonstrated the potential of CliMA Land in tracking the spatial patterns of productivity metrics (GPP) compared to data-driven methods. Similarly, Poulter et al., 2023 recently coupled the LPJ-wsl global DVM and the canopy radiative transfer model PROSAIL. LPJ-PROSAIL can generate global, gridded time series of daily visible to shortwave infrared (VSWIR) spectra (400–2500 nm) taking into account temporal and spatial variability.* |

| | | *Overall, these studies demonstrate that the model couplings (DVM and RTM) are valuable tools for monitoring the development of vegetation activity at the global scale, in strong relation to the carbon cycle and hydrology.* |
|---|---|---|
| • There are a number of current and future missions missing from figure 3, such as EnMAP, EMIT, PACE, and SBG (to name a few). | Thanks for these suggestions; we included future missions (SBG light and heat, EMIT) in Figure 3. In the case of PACE, we decided to not include it as this is an ocean mission and may be less suitable for terrestrial productivity estimations. Regarding EnMAP / PRISMA - these are scientific pre-cursor missions and tasking happens on request. Due to current data take problems in the case of ENMAP, we do not see a huge potential for this mission to acquire time series to obtain vegetation productivity information. Therefore, we decided to not include EnMAP in the figure. The situation for PRISMA may be slightly better, but this mission also aims to prepare future operational missions such as CHIME, and scenes have been acquired often mono-temporally over dedicated sites. | **Section 2.1 / Figure 3:**  |
| • "NBP" is missing from the productivity definition box but is discussed later on. Along these lines, figure 13 doesn't quite make sense to me. Is this saying that NBP is 0.5% of GPP? | Thanks for the good advice! We added NBP to the blue definitions box, to Figure 1 and the corresponding text in that section:

However, Figure 13 is, in our opinion, correct, with GPP > NPP > NEP > NBP as the four main fluxes with increasing time scales but decreasing amounts of stored carbon due to diverse loss processes. Please find also some information in the IPCC report here: https://archive.ipcc.ch/ipccreports/sres/land_use/index.php?idp=24 , stating that *"Compared to the total fluxes between* | **Section 1.1:** *NBP is the net amount of carbon dioxide that is assimilated by an ecosystem over a period of time, after accounting for all losses of carbon dioxide through respiration, decomposition, and other processes. NBP is thus a measure of the overall health and productivity of an ecosystem, and it is an important factor in the global carbon cycle (Prescher et al., 2010, Turner et al., 2007). Input and losses of NBP are on a rather long time scale for natural landscapes and for agriculture it refers to harvest.* **Figure 1:** |

| | | |
|---|---|---|
| | *atmosphere and biosphere, global NBP is comparatively small; NBP for the decade 1989-1998 has been estimated to be 0.7 ± 1.0 Gt C yr -1"* With GPP of 120 Gt C yr-1, the NBP is indeed around 0,5%.

We included a better explanation of Figure 13 in the revised manuscript. |

***Blue box:***
*Net biome productivity (NBP) on a regional scale represents the net change in carbon within ecosystems. It is calculated by adjusting NEP for lateral carbon transfers to neighbouring biomes, which may occur through various processes such as harvest, organic matter export in rivers, or losses from disturbances such as wildfires e.g., Schulze et al., (2021), Prescher et al., (2010).*

**Section 5.5:**
*"Fig. 9 delineates the different levels of productivity which are GPP, NPP, NEP and NBP with respect to their carbon loss processes and flux densities over time. With increasing time scales, the four main fluxes are characterized by decreasing amounts of stored carbon due to diverse loss processes (GPP > NPP > NEP > NBP). Compared to GPP and NPP; NEP and especially NBP are relatively small (IPCC, 2000).* |
| • Figure 7 could be improved. | Since this remark is very open, we were not perfectly sure what exactly we should change. Nonetheless, we improved the figure according to our perception, e.g we removed the double clouds, and added | **Section 4.1., Figure 7:** |

| | arrows to make the position of the three types of validation more clear: |
[Figure]
 |
|---|---|---|
| | | |
| | | |

**Reviewer 2:**

| I would like to start by expressing my appreciation for the effort put into this manuscript. I have read it carefully and I have some feedback that I hope will be helpful in improving the manuscript. My primary concern revolves around the absence of a clear research/review question and the overall lack of structure within the manuscript. To my understanding, the primary objective of this manuscript is to conduct a review of recent advancements in methodologies, sensors, and applications related to remote sensing of vegetation productivity. Vegetation productivity is defined in terms of GPP, NEP, NPP, ABG, crop yield, and harvested wood (as outlined in Table 1). | We would like to thank you for your critical feedback on our paper. We appreciate your time and effort, and we are committed to addressing your concerns.

We agree that the manuscript is lengthy and that more detail is needed in some parts. Therefore, we made some major changes. Mainly, we revised the introduction by preparing a more logical flow and reducing redundancy (see details next comments). Furthermore, we reformulated our objective in **Section 1**, aligned with the advice of the reviewer, into research question:

In response of the question, main research priorities for the identified gaps and challenges are elaborated in more detail in the **Section 6**: Challenges and outlook. See two examples. | **Introduction, research question:**
*What are the state-of-the-art methods for estimating vegetation productivity using remotely sensed TS data streams, and what are the key gaps, challenges, and opportunities for further improvement?*
**Section 6:**
*Our review revealed that multiple gaps, challenges and opportunities exist to accurately estimate vegetation productivity from remotely sensed TS data streams.*
*[..]*
**Section 6.1:**
*Despite this progress, there may still be limitations in terms of the spatial and temporal resolutions for specific objectives and applications of monitoring vegetation productivity trends and processes.*
*[..]* |

| | | |
|---|---|---|
| However, I have a few comments:
The manuscript does not adequately clarify its objectives concerning the review of remotely sensed time series (TS) data in relation to vegetation productivity.
If the aim is to review methods and sensors used to "retrieve/estimate" vegetation productivity from TS data, it appears that the manuscript has not achieved this goal. For example, while multiple sections discuss VIs and RTMs as proxies for productivity, my expectation was that the authors would concentrate on reviewing the methodologies to derive the productivity metrics they defined in table 1 using VIs etc. | Regarding your point: "*...while multiple sections discuss VIs and RTMs as proxies for productivity, my expectation was that the authors would concentrate on reviewing the methodologies to derive the productivity metrics they defined in table 1 using VIs etc…*"
Please note that VIs or RTMs are needed to derive information about productivity metrics as listed in the blue box. It is therefore crucial to concentrate on these methods, which are not always proxies. VIs have been used often as proxies, but RTMs are not direct proxies; they provide traits that can act as proxies or can be further used in process models to derive productivity metrics (like VIs). We made this clearer now (see detailed comments below). | *Bultan et al. (2022), for instance, summarized that plant productivity has been underestimated using DVMs due to missing data from unprecedented extreme events, such as droughts. By providing a long-term perspective and enabling the detection of subtle changes over time, accurate TS data can help here and support more accurate predictions of future trends and impacts.* |
| If the objective is to review time series analyses of vegetation productivity and summarizing the latest "findings" from TS analyses, such as changes in phenology or trends in GPP, it appears that the manuscript has not effectively accomplished this goal. For instance, Section 3.5 discusses tools for time series analyses and preprocessing, yet there are already comprehensive reviews available on these topics as some of them are highlighted in the MS itself. In terms of sensors (Section 2), the manuscript lists several satellites, but to my knowledge, aside from a few exceptions (e.g., MODIS GPP), most of them do not provide productivity metric estimates as defined in this paper. It remains unclear why these sensors are included in the manuscript. Furthermore, VIs and canopy traits | Thanks for the critical feedback. We feel that clarification is needed and re-phrasing in the manuscript: Our main focus lies on the precise remote sensing-based estimation of productivity with consideration of the trend toward the increasing availability of higher spatial resolution EO data. Global change is resulting in a landscape, which is more fragmented, scattered and characterized by small patterns. One example is the upcoming trend of agroforestry to make agriculture more resilient. As a consequence, the analysis of productivity needs to integrate high spatial resolution remote sensing data and we preferred to focus more on the spatial scale than on the minimum number of time steps.
See our text changes following the research question:
One of the main keywords of our systematic literature review was "time series." We did not initially define a minimum number of consecutive observations for inclusion in the review. Unlike other | **Introduction, following the research question:**
*To address this question, our main emphasis is on the precise EO-based estimation of productivity with consideration of the trend towards the increasing availability of higher spatial resolution EO data. Global change is resulting in a landscape that is more fragmented, scattered, and characterized by small-scale patterns. One example is the upcoming trend of agroforestry to make agriculture more resilient. As a consequence, the analysis of productivity needs to integrate high-resolution EO data.*
*Hence, we will focus on the literature that uses remotely sensed optical TS and derived proxies for quantifying productivity, with a greater emphasis on the spatial scale than on the minimum number of time steps.*

**Section 5.1:**
*Unlike other studies, which defined a time series as* |

| | | |
|---|---|---|
| derived from these sensors are considered loose proxies for vegetation productivity, and their time series analyses may not necessarily align with time series of productivity metrics. | papers, which define a time series as consisting of a minimum of several observations, we included studies with a minimum of two images without an upper limit. This allowed us to include studies that have traditionally been labeled under the topic of change detection analysis.

We chose to do this for two reasons. First, we believe that the minimum number of observations in a time series is arbitrary, and we wanted to take a more comprehensive approach to examining the aspect of time. Second, the number of studies using long time series consisting of tens to hundreds of high-resolution (10-30 m pixel size) images is relatively small. If we had only looked at long time series, we would have excluded many studies that observe productivity from Landsat and Sentinel-2 satellites. | *consisting of a minimum of several observations, we included studies with a minimum of two images without an upper limit. This allowed us to include studies that have traditionally been labeled under the topic of change detection analysis. We chose to do this for two reasons. First, we believe that the minimum number of observations in a time series is arbitrary, and we wanted to take a more comprehensive approach to examining the aspect of time. Second, the number of studies using long TS consisting of high-resolution (10-30 m pixel size) images is relatively small. Considering only long TS data streams would have excluded many studies that observe productivity from Landsat and Sentinel-2 satellites.* |
| I am not trying to impose my view on how the review should be structured, all I'm asking for is more clear objectives:
1)
The manuscript is unnecessarily lengthy and mostly provides general information without delving deeply into each subject. The introduction (Section 1) is overly general, and much of the information is repeated later in the text. I would recommend revising it to clearly outline the main objectives and rationale for the need for a new review.

Section 1.1 requires some refinement, particularly regarding photosynthesis, a key component of productivity. Notably, the authors seem to have overlooked important literature, including reviews, on | Thanks for your suggestions! The overall aim of **Section 1** was to give an overview of the different concepts, sensors, and methods, which are then explained more in detail in the following sections. We agree with you that some information is redundant. Therefore we shortened the chapter, deleting redundant sentences and shifting some parts to the other chapters:

Specifically, we refined **Section 1.1,** and moved some sentences about photosynthesis from another part to here. Among others, Ryu et al., 2019 have been included in **Sections 1. and 1.1**.:
Besides, we included the definition of net biome production in this section and the blue box (see reviewer 1 comments).

We removed **Section 1.2.** We deleted the part about specific sensors, as is exhaustively discussed in | **Section 1:**
*Vegetation productivity, the rate at which solar energy is converted into biomass through photosynthesis, is the origin of all fuel, fiber, and food by which humanity and many other species live, and should therefore be closely monitored. The total amount of photosynthesis on Earth defines the planetary boundary of production, which is a measure of how much of the planet's productivity humans have appropriated(Ryu et al., 2019).*

**Section 1.1:**
*Vegetation productivity is controlled by two processes; (i) the assimilation of CO2 substrate through photosynthesis (source activity) and (ii) tissue growth from the accumulated carbohydrates into stored biomass (sink activity) (Korner et al., 2015). Plant photosynthesis is driven by incoming photosynthetic active radiation, CO2 concentrations, temperature,* |

| | | |
|---|---|---|
| this topic (e.g., Ryu et al., 2019).

Section 1.2 is disorganized, with information being repeated in other sections.

In Section 1.3, it's essential to emphasize that we estimate productivity using remote sensing data rather than measuring it directly. Additionally, the authors have listed VIs and some other variables as productivity metrics, which does not align with my expectation based on Table 1. | **Section 2**. Then we moved the part about phenology to **Section 1**., and the paragraph about gap-filling to **Section 3.1.3** to avoid repetitions.

In **Section 1.3 (actual Section 1.2)**, we emphasized that we estimate productivity using remote sensing data rather than measuring it directly. See some exemplary sentences:

We also clarified that VIs and other variables are not productivity metrics, but are used for deriving productivity metrics: | *and water and nutrient availability, e.g. Ryu et al., (2019).*

**(new) Section 1.2:**
*The presence of strong absorption features in optical wavelengths, which relate to biochemical properties such as pigment and water content, has led to a large body of research using optical sensors to monitor vegetation productivity, mitigating the need for direct measurements (Boisvenue et al. 2016, Brinkmann et al. 2011, Cai et al. 2021, Dusseux et al. 2022, Erasmi et al. 2021, Hill et al. 2003).*
*[..]*
*Traditionally, spectral vegetation indices (VIs) have been used to derive plant productivity metrics..*
*[..]*
*"Data-driven RS-based approaches may include the establishment of statistical relationships through empirical approaches or, more recently, with machine learning (ML) algorithms (see review by Liao et al. (2023).*
*[...]*
*Over the last decade, solar-induced fluorescence (SIF) from space measurements has become increasingly popular,...*
*[...]*
*Ardo (2015) suggested that the integration of the realistic processes simulated by DVMs with the high-resolution RS observational may support more accurate productivity metrics estimation. These approaches are discussed in more detail in Section 3.* |
| Section 2.4 lists platforms that are merely web applications facilitating data download and processing from other remote sensing platforms. These platforms serve different purposes, and it's unclear why they are presented here. | Agreed. **Section 2.4** is removed. | |

| | | |
|---|---|---|
| Overall, Section 2 offers general information on sensors and platforms and needs revision to align more consistently with the productivity metrics. | Thanks for the good advice. We related the sensor/platform usage more directly to the estimation of productivity metrics, such as GPP and yield in the following parts:

In several parts of the chapter, there was already a reference to productivity metrics, such as with the Copernicus pan-European High-Resolution Vegetation Phenology and Productivity product suite, or exemplarily studies like Reyes-Munoz et al (2022), who derived traits for potential usage in productivity metrics derivation (**2.1**).

With the new text and by excluding **Section 2.4,** we feel that **Section 2** now more directly relates the sensors to productivity metrics. | **Section 2:**
*This increase in the abundance of EO data has contributed to the establishment of consistent global databases with quality-checked optical data, which can be used to estimate vegetation productivity metrics, such as GPP, NPP, AGB, yield, among others (see also blue box) at almost any spatial and temporal scales (Kuenzer et al., 2015).*
*[..]*
*This may explain why we could identify only a few studies that employed piloted aircraft to acquire optical TS for the estimation of vegetation productivity metrics, such as Damm et al., (2015) focusing on SIF. In this study, the authors conducted a thorough evaluation of the correlation between far-red SIF measured at 760 nm and GPP across three ecosystems, namely perennial grassland, cropland, and mixed temperate forest, using Airborne Prism EXperiment (APEX) TS sensor data. The authors concluded that remote sensing of SIF more consistently correlated to GPP than conventional greenness-based remote sensing indices.*
*[..]*
*Also, UAVs offer the necessary flexibility to sample diurnal cycles, which are relevant to capturing trends in productivity.*
*[..]*
*A recent phenotyping UAV study, however, collected UAV data from a soybean field trial at unprecedented temporal resolution (Borra et al., 2020), which allowed fitting growth curves with high accuracy (90%) to derive relevant traits but also seed yield.* |
| 2. Please refer to my previous comments on VIs and traits in Sections 3.1.1 and 3.1.2. | Thanks indeed we needed to adjust these chapters. See some of the improved sentences: | **Section 3.1.1:**
*VIs are widely applied methods for monitoring trends and deriving plant productivity metrics, such as GPP..* |

| | | |
|---|---|---|
| | Please also see our response to your next comment for section 3, where we strongly emphasized the role of VIs and traits as means /inputs into specific methods such as phenology or process models to derive productivity metrics (such as yield / GPP). For instance, we removed the statement about proxies. | *[..]*
*Multiple studies have explored TS of NDVI and EVI with direct linkages to vegetation productivity metrics, such as GPP…*

**Section 3.1.2:**
*To obtain productivity metrics, TS data streams of the traits have been integrated into various GPP assimilation schemes (e.g., Jung et al., 2007, Xie et al., 2019, Chen et al., 2022)*
*[..]*
*An overview of widely used quantitative traits in TS processing available from RTM inversion, and their relationship to potential vegetation productivity is given in Table 1. These traits can be further used within defined methodologies to derive productivity metrics given in the blue box, such as GPP.* |
| | | |
| 3. Sections 3.1.3 to 4.2 present various time series studies, toolboxes, and variables ranging from VIs to phenology and GPP. The main objectives of these sections are unclear and appear to require revision for clarity and focus (please refer to point #1). | The aim of **Sections 3.1.3 - 3.5** is to provide a methodological overview of productivity metrics estimation trend analysis (**3.2**) using land surface phenology (**3.3**), and more advanced models (**3.4**.), and finally toolboxes (**3.5**).

Specifically, **Section 3.1.3** introduces challenges of time series analysis like data gaps and solutions, needed to use TS for deriving productivity metrics (via trends, LSP, models..). As it seems that this logic is not clear to the reader, we introduced the overall section aim at the beginning of **Section 3**.

Throughout these sections, the relevance of identified methods to measure productivity has been highlighted in the text. See also further changes in the text: | **Section 3:**
*This section introduces several methods for deriving productivity metrics from remotely sensed TS data, including trend analysis, land surface phenology, and process models. Each method has its own strengths and weaknesses, and the best approach to use will depend on the specific application. The final sub-chapter of this chapter will introduce a variety of toolboxes that can be used to process and analyze remotely sensed TS data and derive productivity metrics.*
*By providing a comprehensive overview of the different methods and tools available, this chapter aims to help researchers and practitioners select the best approach to deriving productivity metrics from remotely sensed TS data for their specific needs.*

**Section 3.2:** |

| | Note that in **Section 3.4** we cited some more relevant studies (as suggested by reviewer 1) that approach the Digital Twin concept by combining DVMs with RTMs (forward mode) to estimate productivity metrics from time series (using simulations and RS measurements).

In **Section 3.5** we made clearer that the toolboxes are mainly to derive traits which can be further used within the presented methods to derive productivity metrics.

**Section 4** introduces validation strategies since validation is critical in understanding the accuracy and reliability of estimated quantities - hence productivity metrics. Besides, we adapted Figure 7 (see reviewer 1 response). | *Although all three VIs produced similar trends in SOS, a pronounced land-cover dependence was observed, with PPI-SOS outperforming the other two spectral indices in approximating vegetation productivity, i.e. GPP.*

**Section 3.3:**
*For instance, Wood et al. (2021) used three decades of AVHRR data over the U.S. Northwestern Plains to study the impact of climate change and agricultural management on phenology. They concluded that climate factors such as precipitation and temperature can have a significant impact on productivity, but other factors such as soil nutrients, disturbance, and management practices also play a role.*
*[..]*
*The concept of LSP also has its drawbacks. Apart from the influence of the smoothing technique and the method used to extract the LSP metrics, Helman et al., (2018) stressed that changes in vegetation species composition rather than phenological transitions could produce a false-positive signal in LSP. Moreover, LSP metrics show high sensitivity to the frequency and temporal coverage of observations as well as cloud contamination, which can affect the estimation of productivity metrics (Younes et al 2021).*

**Section 3.4:**
*For instance, The Breathing Earth System Simulator (BESS) model (Ryu et al., 2011, Jiang et al 2016) couples atmosphere and canopy processes, two-leaf photosynthesis, and energy balance, to provide evapotranspiration and GPP.*
*[..]*
*Shiklomanov et al., (2021), for instance, coupled three existing models, namely the Ecosystem Demography model version 2 (ED2, (Medvigy et al. (2009) PROSPECT-5 (Feret et al., 2008) and a simple soil* |
|---|---|---|

| | | *reflectance model to the EDR model. Their model predicts the full range of high-spectral-resolution surface reflectance, which is dependent on the current state of the ED2 model. Another relevant example is provided by (Wang et al., 2023) with Climate Modeling Alliance (CliMA) Land, which is able to simulate productivity metrics such as GPP, transpiration, as well as canopy reflectance and fluorescence spectra that can be observed by satellites in a high temporal resolution. The authors demonstrated the potential of CliMA Land in tracking the spatial patterns of productivity metrics (GPP) compared to data-driven methods. Similarly, Poulter et al. (2023) recently coupled the LPJ-wsl global DVM and the canopy radiative transfer model PROSAIL. LPJ-PROSAIL can generate global, gridded time series of daily visible to shortwave infrared (VSWIR) spectra (400–2500 nm) taking into account temporal and spatial variability. Overall, these studies demonstrate that the model couplings (DVM and RTM) are valuable tools for monitoring the development of vegetation activity at the global scale, in strong relation to the carbon cycle and hydrology.*

**Section 3.5:**
*A variety of sophisticated software packages have been developed to facilitate the processing and analysis of large image TS and ultimately provide key information about vegetation dynamics and ultimately about productivity metrics.*

**Table 2 caption:**
*Toolboxes recommended and used for converting remotely sensed TS into gap-filled VI and vegetation trait products, and to derive LSP metrics and trends, which all can be ultimately used for estimating productivity metrics.* |
|---|---|---|

| | | |
|---|---|---|
| | | **Section 4:**
*Ultimately, validation is essential for ensuring that remotely sensed TS data can be used to accurately estimate GPP, NPP, and other vegetation productivity metrics.* |
| 4. Section 5.1 is one of the strongest parts of MS. It interesting to know that many studies refer to VIs as productivity (Figure 9). It would be nice to link this section to GPP, AGB as authors defined them as productivity.

The application section is very general. I think it needs revision to make it more focused on the objectives of the manuscript. | Thanks for your positive assessment. We made some changes in section 5.1. Besides moving the PRISMA overview figure to the appendix (according to reviewer 1 suggestions), we included a figure (Fig. 8) presenting the histogram with the number of time steps used by the reviewed studies.

Furthermore, We linked **Sections 5.3/5.4** more explicitly to AGB and GPP, as these subsections go more into detail about practical applications. For **Section 5.1** we kept the general description related to methods. | **Section 5.1.**
*Fig. 8 shows the number of published papers per number of explored TS observations.*
*Note that the x-axis starts with "2". There is a skewed normal distribution with a median of 227 temporal observations and a long tail towards a higher number of observations. The 75th percentile is reached at 786 observations.*

[Figure]

**Section 5.3.:**
*The review by Nguyen et al (2020) stated that innovative Landsat-based approaches for estimating forest AGB dynamics across space and time have been developed in recent years. Methods have become more advanced and robust over time. For instance, Landsat data can be used to fill in missing data points in AGB maps, which can improve the overall quality of the maps and make them more useful for applications such as carbon accounting and forest monitoring. Landsat data have been also used* |

| | | |
|---|---|---|
| | | *to estimate AGB over large areas and long time periods, even in areas where there is limited field data. Furthermore, recovery metrics can be used to improve the accuracy of AGB models since Landsat data can provide information about the dynamics of forests over time, which is not always captured by traditional AGB models.*
*[..]*
*Climate change is also increasing the frequency of biotic disturbances like insect outbreaks (Senf et al., 2017, Olsson et al., 2017). Insect outbreaks in forests have a significant impact on productivity by defoliating trees and changing the structure of the forest.*

**Section 5.4.:**
*For natural ecosystems, many studies assessed spatial and temporal trends in vegetation productivity for specific ecosystem types, often based on phenology indicators derived from TS of spectral VIs or by integrating those in DVMs.*
*[..]*
*The MODIS GPP algorithms were also used in GPP estimation studies (Feagin et al., 2020) and for a comparison with a LUE-based DVM (Liu et al., 2011). Studies also compared the results of estimating GPP using MODIS and S2 TS data (Cai et al., 2021).*
*[..]*
*A few studies explored MODIS TS and EC flux tower data (e.g., Kang et al., 2018, Wang et al., 2021). Wang et al. (2021), for instance, used TS of the MOD17A3 annual NPP product to reveal spatial and temporal trends of NPP in China, among others.* |

We sincerely thank both the reviewers and the editor for their justified comments and valuable advice! We hope that with the changes performed, we can adequately address your concerns and essentially improve our manuscript.

Kind regards, Katja Berger, Lammert Kooistra & Co-authors